# LLaVA-Mini: Efficient Image and Video Large Multimodal Models with One Vision Token

**Shaolei Zhang**[1,3], **Qingkai Fang**[1,3], **Zhe Yang**[1,3], **Yang Feng**[1,2,3*]

[1]Key Laboratory of Intelligent Information Processing,
  Institute of Computing Technology, Chinese Academy of Sciences (ICT/CAS)
[2]Key Laboratory of AI Safety, Chinese Academy of Sciences
[3]University of Chinese Academy of Sciences, Beijing, China
  `zhangshaolei20z@ict.ac.cn, fengyang@ict.ac.cn`

## Abstract

The advent of real-time large multimodal models (LMMs) like GPT-4o has sparked considerable interest in efficient LMMs. LMM frameworks typically encode visual inputs into vision tokens (continuous representations) and integrate them and textual instructions into the context of large language models (LLMs), where large-scale parameters and numerous context tokens (predominantly vision tokens) result in substantial computational overhead. Previous efforts towards efficient LMMs always focus on replacing the LLM backbone with smaller models, while neglecting the crucial issue of token quantity. In this paper, we introduce LLaVA-Mini, an efficient LMM with minimal vision tokens. To achieve a high compression ratio of vision tokens while preserving visual information, we first analyze how LMMs understand vision tokens and find that most vision tokens only play a crucial role in the early layers of LLM backbone, where they mainly fuse visual information into text tokens. Building on this finding, LLaVA-Mini introduces modality pre-fusion to fuse visual information into text tokens in advance, thereby facilitating the extreme compression of vision tokens fed to LLM backbone into one token. LLaVA-Mini is a unified large multimodal model that can support the understanding of images, high-resolution images, and videos in an efficient manner. Experiments across 11 image-based and 7 video-based benchmarks demonstrate that LLaVA-Mini outperforms LLaVA-v1.5 with just 1 vision token instead of 576. Efficiency analyses reveal that LLaVA-Mini can reduce FLOPs by 77%, deliver low-latency responses within 40 milliseconds, and process over 10,000 frames of video on the GPU hardware with 24GB of memory.[1]

## 1 Introduction

Large multimodal models (LMMs), such as GPT-4o (OpenAI, 2024), equip large language models (LLMs) (OpenAI, 2022; 2023) with the ability to understand visual information, exhibiting a common trend toward low-latency responses to enable real-time multimodal interactions. Recently, the most widely adopted LMMs (Liu et al., 2023b; 2024a; Zhu et al., 2024), exemplified by the LLaVA series (Liu et al., 2023b), involves embedding image patches into vision tokens through a vision encoder (Radford et al., 2021) and incorporating them into the LLM's context to facilitate visual information comprehension, leading to strong performance in image and video understanding.

However, the substantial computational costs of LMMs present ongoing challenges. Unlike LLMs (Touvron et al., 2023a;b; Dubey et al., 2024), which only process textual inputs, LMMs must incorporate a large number of additional vision tokens into the LLM's context to represent visual information (Liu et al., 2023b), significantly increasing computational complexity. For instance, in the widely used vision encoder CLIP ViT-L/336px, a single image is encoded into $24 \times 24 = 576$ vision tokens (Radford et al., 2021), where integrating such a large number of vision tokens into

---

[*]Corresponding author: Yang Feng.

[1]Code: `https://github.com/ictnlp/LLaVA-Mini`; Model: `https://huggingface.co/ICTNLP/llava-mini-llama-3.1-8b`

the context of parameter-heavy LLM results in significant computational overhead and higher inference latency. This issue becomes even more pronounced in high-resolution image modeling (which requires more vision tokens per image) (Liu et al., 2024b) or video processing (which involves processing more images) (Maaz et al., 2024; Lin et al., 2023a). Therefore, developing efficient LLMs is essential for achieving GPT-4o-like low-latency multimodal interactions.

The computational demands of LMMs are primarily driven by model scale and the number of tokens in the input context. Existing approaches to improving LMM efficiency typically focus on model downsizing (Chu et al., 2023; 2024; Yuan et al., 2024a; Zhou et al., 2024a) or quantization techniques (Yuan et al., 2024b), but often overlook another critical avenue: reducing the number of vision tokens to shorten the input context. Some token reduction methods rely on predefined rules to reduce the number of tokens output by the vision encoder (Bolya et al., 2023; Shang et al., 2024; Li et al., 2024e; Ye et al., 2024d; Hu et al., 2024), which leads to the loss of visual information and inevitably results in performance degradation (Wang et al., 2024; Fan et al., 2024).

In this paper, we aim to develop efficient LMMs by minimizing the number of vision tokens while maintaining comparable performance. To this end, we begin by exploring a foundational question: *How does the LMM (particularly the LLaVA architecture) understand vision tokens?* Through layer-wise analysis (refer to Sec.3), we observe that the importance of vision tokens changes across different layers of LLM. In the early layers, vision tokens play a crucial role, receiving considerable attention from the following text tokens (e.g., user input instructions and responses). However, as the layers deepen, the attention devoted to vision tokens decreases sharply, with most attention shifting towards the input instructions. Notably, even when we entirely remove vision tokens in some later layers, LMM keeps certain visual understanding capabilities. This finding suggests that vision tokens are more critical in early layers, where text tokens fuse visual information from vision tokens.

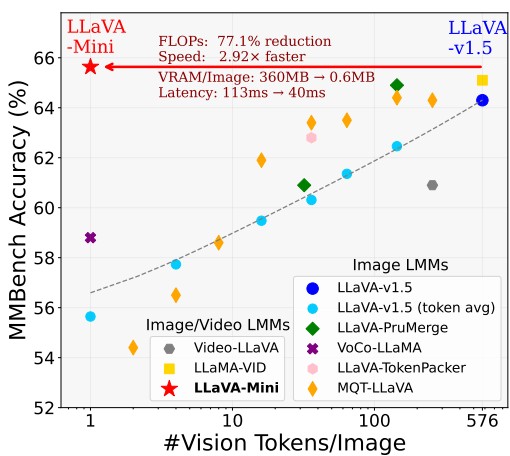

Figure 1: LLaVA-Mini achieves comparable performance to LLaVA-v1.5 using only 1 vision token instead of 576, yielding efficient computation, lower latency, and reduced VRAM usage.

Based on this finding, if the fusion process can be shifted from the early layers of LLM to perform before LLM, we can significantly reduce the number of vision tokens fed into the LLM without sacrificing performance. Along with this idea, we propose *LLaVA-Mini*, an efficient and high-quality LMM with minimal vision tokens. LLaVA-Mini introduces a modality pre-fusion module before LLM to fuse visual information into the instruction text in advance, and employs a compression module to highly compress the vision tokens before inputting them into LLM, thereby enhancing efficiency while preserving high-quality visual understanding. Under extreme settings, LLaVA-Mini requires only one vision token per image fed into LLM backbone, offering significant advantages in inference time and memory consumption for high-resolution image and long video processing.

Experiments across a wide range of 11 image-based and 7 video-based understanding benchmarks show that LLaVA-Mini achieves performance comparable to LLaVA-v1.5 (Liu et al., 2023b) while using only 1 vision token instead of 576 (compression rate of 0.17%). With minimal vision tokens, LLaVA-Mini offers substantial benefits in terms of computational efficiency (77% FLOPs reduction) and lowering GPU memory usage (360 MB → 0.6 MB per image), as shown in Figure 1. As a result, LLaVA-Mini decreases inference latency of image understanding from 100 ms to 40 ms and also enables the processing of long videos exceeding 10,000 frames (over 3 hours) on an NVIDIA RTX 3090 with 24GB of memory, paving the way for low-latency multimodal interactions.

## 2 RELATED WORK

As Large multimodal models (LMMs) are increasingly deployed in real-time applications (OpenAI, 2024), enhancing their efficiency has become a critical concern. Recent efforts focus on either

reducing the model size or the number of tokens that fed into LMM. To reduce LMM's model size, previous methods directly replace the LLM backbone with a smaller one (Chu et al., 2023; 2024; Yuan et al., 2024a; Zhou et al., 2024a), while directly reducing the parameter scale can impact the LLM backbone's capabilities, resulting in performance declines in visual tasks (Shang et al., 2024).

Another efficiency determinant for LMMs is the context length provided to the LLM backbone, including vision and text tokens. In practice, the number of vision tokens can be substantial, particularly when processing high-resolution images and videos. For image-based LMMs, token merging (Bolya et al., 2023), PruMerge (Shang et al., 2024), and TokenPacker (Li et al., 2024e) aggregate vision tokens based on similarity. Qwen-VL (Bai et al., 2023) and MQT-LLaVA (Hu et al., 2024) utilize Q-former (Li et al., 2023a) to compress vision tokens into a fixed length. However, directly reducing vision tokens inevitably results in the loss of visual information (Fan et al., 2024).

For video-based LMMs, Video-ChatGPT (Maaz et al., 2024), VideoChat (Li et al., 2024c), Video-LLaVA (Lin et al., 2023a), and Video-LLaMA (Zhang et al., 2023), select a fixed number of frames from videos of varying lengths. MovieChat (Song et al., 2024a) applies memory techniques to condense videos into a fixed-length representation. Such frame selection or merging methods may lose some key frames or misunderstand the temporal information of the video (Zhou et al., 2024b).

Previous methods have primarily focused on token reduction on the vision encoder. LLaVA-Mini takes this a step further by exploring how vision tokens and text tokens interact within the LLM backbone, and accordingly introduces a modality pre-fusion module, enabling an extreme compression of vision tokens (1 vision token fed into LLM) while achieving comparable performance.

## 3 How Does LLaVA Understand Vision Tokens?

To compress visual tokens while preserving visual understanding, we sought to figure out how LMMs understand visual tokens. Given the complexity of this issue, our preliminary analysis concentrated on the LLaVA architecture (Liu et al., 2023b), focusing on the role of visual tokens (particularly their quantity) in LMMs from an attention-based perspective (Xiao et al., 2024).

### 3.1 LLaVA Architecture

LLaVA (Large Language and Vision Assistant) (Liu et al., 2023b) is an advanced multimodal architecture that integrates vision and language processing capabilities. Building upon vision Transformers (ViT) (Dosovitskiy et al., 2021) for visual inputs and LLMs for text, LLaVA can generate language response $\mathbf{X}^a$ based on the given language instruction $\mathbf{X}^q$ and visual inputs $\mathbf{X}^v$.

Typically, a pre-trained CLIP ViT-L/14 (Radford et al., 2021) and a projection layer are employed to encode the visual inputs $\mathbf{X}^v$ into vision tokens (i.e., continuous representations) $\mathbf{H}^v$. Then, vision tokens $\mathbf{H}^v$ and language instruction's embedding $\mathbf{H}^q$ are fed into an LLM, such as Vicuna (Chiang et al., 2023) or Mistral, to generate the response $\mathbf{X}^a$. In practice, vision tokens are often inserted into the middle of the language instruction, so the inputs of LLM can be formally represented as:

$$\left\langle H_1^q, \cdots, H_k^q, H_1^v, \cdots, H_{l_v}^v, H_{k+1}^q, \cdots, H_{l_q}^q \right\rangle, \tag{1}$$

where $l_v$ and $l_q$ denote the lengths of the vision tokens and language instruction, respectively. For instance, in LLaVA-v1.5, the system prompts are positioned before the image (i.e., $H_1^q, \cdots, H_k^q$), while the user inputs follow the image (i.e., $H_{k+1}^q, \cdots, H_{l_q}^q$) (Liu et al., 2023b).

### 3.2 Preliminary Analyses

We begin by analyzing the significance of visual tokens in LMMs to guide the strategies for compressing vision tokens. Specifically, we evaluate the importance of visual tokens at each layer of LMMs from an attention-based perspective. Our analysis encompasses several LMMs, including LLaVA-v1.5-Vicuna-7B, LLaVA-v1.5-Vicuna-13B, LLaVA-v1.6-Mistral-7B, and LLaVA-NeXT-Vicuna-7B (Liu et al., 2023b; 2024b), to identify common characteristics across models of varying sizes and training datasets. Appendix A gives the formal expression of the preliminary analyses.

**Vision Tokens are More Important in Early Layers** To find out which layers in LMM the vision tokens play a more important role, we measure the attention weights assigned to different token

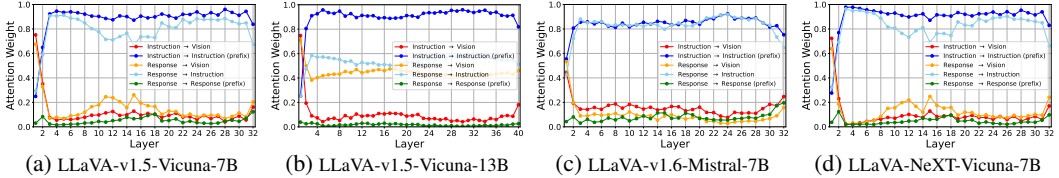

Figure 2: Layer-wise variation of attention weights assigned to different types of tokens (including instruction, vision, and response) in LMMs. "A→B" means the attention weights from A to B.

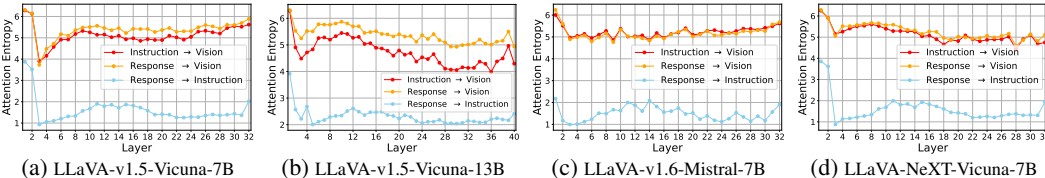

Figure 3: Attention entropy assigned to different types of tokens across different layers in LMMs.

types (including instruction, vision, and response) at each layer. As shown in Figure 2, Visual tokens receive more attention in the earlier layers, but this attention sharply decreases in the deeper layers, with over 80% of the attention being directed towards instruction tokens. This finding is consistent with the previous conclusion (Chen et al., 2024). This change in attention suggests that vision tokens play a central role in the early layers, with the instruction tokens seeking relevant visual information from vision tokens through attention mechanisms. In the later layers, the model relies more on instructions that have already fused the visual data to generate responses.

**Most Vision Tokens are Focused in Early Layers** To further assess the importance of individual visual tokens, we calculate the entropy of the attention distribution at each layer. As shown in Figure 3, we find that the entropy of attention toward visual tokens is much higher in the earlier layers, indicating that most visual tokens are evenly attended to in the early layers.



Figure 4: Attention visualization at different layers in LLaVA-v1.5 (color bar: logarithmic scale).

To intuitively illustrate the layer-wise variations in the importance of visual tokens, Figure 4 visualizes the attention distribution across each layer of LLaVA-v1.5. Almost all visual tokens receive broader attention in the early layers, while only some visual tokens are focused in the later layers. These observations suggest that all visual tokens are crucial in the early layers, and reducing their quantity inevitably results in a loss of visual information. This explains why previous methods of direct token reduction will compromise visual understanding capabilities (Shang et al., 2024; Ye et al., 2024d; Hu et al., 2024).

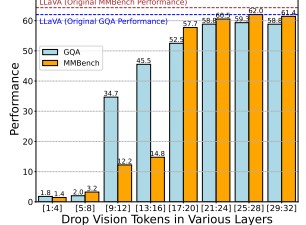

Figure 5: Performance of LLaVA-v1.5 when removing all vision tokens in various layers of LMM.

To further substantiate our finding that visual tokens are particularly critical in the early layers, we evaluated the visual understanding ability of LMMs when visual tokens were dropped at different layers. Specifically, we measured the performance of LLaVA-v1.5 on the GQA (Hudson & Manning, 2019) and MMBench (Liu et al., 2024c), with visual tokens being dropped at layers 1-4, 5-8, ... , 29-32, respectively. As shown in Figure 5, removing visual tokens in the early layers leads to a complete loss of visual understanding ability, while removing tokens in the higher layers has a minimal effect, with the model retaining much of its original performance. In conclusion, our analyses and ablation study reveal that vision tokens play a crucial role in the early layers of LLaVA, where text tokens fuse visual information from the vision tokens at this stage. This insight can inform our strategies for compressing vision tokens.

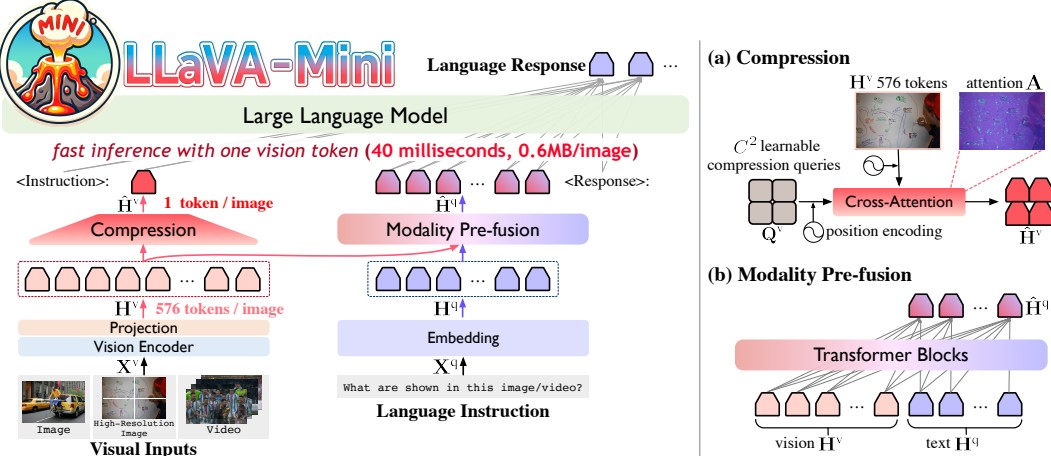

Figure 6: Architecture of LLaVA-Mini. **Left**: LLaVA-Mini represents each image with one vision token. **Right**: Detailed view of the proposed query-based compression and modality pre-fusion.

# 4 LLAVA-MINI

We introduce LLaVA-Mini, an efficient large multimodal model with minimal vision tokens. Like previous work, LLaVA-Mini uses a vision encoder to encode an image into several vision tokens. To enhance the efficiency, LLaVA-Mini significantly reduces the number of vision tokens fed into LLM backbone through a compression module. To retain visual information during compression, based on previous findings that vision tokens play a crucial role in the early layers for fusing visual information, LLaVA-Mini introduces a modality pre-fusion module before the LLM backbone to fuse the visual information into the text tokens. The details of LLaVA-Mini are as follows.

## 4.1 ARCHITECTURE

The architecture of LLaVA-Mini is illustrated in Figure 6. For the visual inputs $\mathbf{X}^{\mathrm{v}}$, a pre-trained CLIP vision encoder (Radford et al., 2021) is employed to extract visual features from each image. These features are then mapped into the word embedding space via a projection layer, producing vision tokens $\mathbf{H}^{\mathrm{v}} \in \mathbb{R}^{N^2 \times d_h}$, where $N^2$ is the number of vision tokens and $d_h$ is the LLM's embedding dimension. For the language instruction $\mathbf{X}^{\mathrm{q}}$, LLM's embedding layer is used to generate text token representations $\mathbf{H}^{\mathrm{q}} \in \mathbb{R}^{l_q \times d_h}$, where $l_q$ is the number of text tokens.

**Vision Token Compression**  To enhance the efficiency of LMMs, LLaVA-Mini reduces the number of vision tokens fed into the LLM backbone by utilizing a query-based compression module. To learn compression of the vision tokens, LLaVA-Mini introduces $C \times C$ learnable compression queries $\mathbf{Q}^{\mathrm{v}}$. These queries interact with all vision tokens $\mathbf{H}^{\mathrm{v}}$ through cross-attention (Li et al., 2023a; Ye et al., 2024a), selectively extracting the important visual information to produce $C \times C$ compressed vision tokens $\hat{\mathbf{H}}^{\mathrm{v}} \in \mathbb{R}^{C^2 \times d_h}$. To preserve the spatial information in the image during compression, we introduce a 2D sinusoidal positional encoding $PE(\cdot)$ (He et al., 2021) on the learnable queries and original vision tokens. Formally, the compression can be expressed as:

$$\hat{\mathbf{H}}^{\mathrm{v}} = \mathbf{A} \cdot \mathbf{H}^{\mathrm{v}}, \quad \text{where} \quad \mathbf{A} = \text{Softmax}\left( (\mathbf{Q}^{\mathrm{v}} + PE(\mathbf{Q}^{\mathrm{v}})) \cdot (\mathbf{H}^{\mathrm{v}} + PE(\mathbf{H}^{\mathrm{v}}))^{\top} \right), \quad (2)$$

where $\mathbf{A} \in \mathbb{R}^{C^2 \times N^2}$ is the similarity and $\hat{\mathbf{H}}^{\mathrm{v}}$ are $C \times C$ compressed vision tokens.

**Modality Pre-fusion**  The compression of vision tokens inevitably results in some loss of visual information. To retain as much visual information during compression as possible, LLaVA-Mini introduces a modality pre-fusion before the LLM backbone, enabling text tokens to fuse relevant visual information from all vision tokens in advance. Based on our previous observations, where this fusion stage occurs implicitly within the early layers of the LLM, the modality pre-fusion module $f(\cdot)$ consists of $N_{fusion}$ Transformer blocks (Vaswani et al., 2017), where each Transformer block shares the same structure and hyperparameters with LLM backbone. Vision tokens $\mathbf{H}^{\mathrm{v}}$ and text

tokens $\mathbf{H}^q$ are concatenated and fed into the pre-fusion module, and the outputs corresponding to the text tokens are then extracted as fusion tokens, expressed as:

$$\hat{\mathbf{H}}^q = f\left(\text{Concat}\left(\mathbf{H}^v, \mathbf{H}^q\right)\right)\left[-l_q :\right], \tag{3}$$

where $\hat{\mathbf{H}}^q \in \mathbb{R}^{l_q \times d_h}$ are fusion tokens of text representations with related visual information.

Finally, the compressed vision tokens $\hat{\mathbf{H}}^v$ and fusion tokens $\hat{\mathbf{H}}^q$ of text representations with related visual information ($C^2 + l_q$ tokens in total) are fed to LLM together to generate the response.

## 4.2 HIGH-RESOLUTION IMAGE AND VIDEO MODELING

LLaVA-Mini uses minimal vision tokens to represent visual information, making it possible to handle high-resolution images and videos much more efficiently.

**High-Resolution Image** The resolution of LMM is typically determined by the vision encoder, such as CLIP's ViT-L, which encodes at a resolution of 336*336 pixels. To perceive images at a higher resolution, we divide each image into four sub-images by splitting it horizontally and vertically into two parts (Liu et al., 2024b). Each of these sub-images is processed by the vision encoder and projection individually, yielding $N^2 \times 4$ vision tokens with a high resolution of 672*672 pixels. The proposed compression module is then employed to reduce these $N^2 \times 4$ vision tokens into $C^2$ compressed vision tokens $\hat{\mathbf{H}}^v$. The modality pre-fusion module takes the 4 sub-images ($N^2 \times 4$ vision tokens), the original image ($N^2$ vision tokens), and the language instruction ($l_q$ text tokens) as inputs, and then generates $l_q$ fusion tokens $\hat{\mathbf{H}}^q$ with richer global and local visual information. Finally, the number of tokens input to the LLM is $C^2 + l_q$. Note that when handling high-resolution images, $C$ is set slightly higher than in standard-resolution settings to preserve more details.

**Video** When handling videos, LMMs often extract multiple frames from the video (Li et al., 2023b), which incurs significant computational costs. For instance, in the case of LLaVA-v1.5, extracting frames at a rate of 1 frame per second (fps) from an 8-second video results in $576 \times 8 = 4608$ vision tokens, leading to substantial VRAM usage. LLaVA-Mini can represent each image with minimal vision tokens, providing a significant advantage in processing long videos. For a video consisting of $M$ frames, LLaVA-Mini processes each frame individually, generating $C^2$ vision tokens and $l_q$ fusion tokens per frame. $C^2$ vision tokens from each of $M$ frames are sequentially concatenated to yield a total of $M \times C^2$ vision tokens, i.e., $\hat{\mathbf{H}}^v$. Then, $l_q$ fusion tokens corresponding to $M$ frames are aggregated through pooling operation to generate the video's fusion tokens $\hat{\mathbf{H}}^q$. As a result, the number of tokens fed to the LLM is reduced from $MN^2 + l_q$ to $MC^2 + l_q$ for a video of $M$ frames.

## 4.3 TRAINING

LLaVA-Mini follows the same training process as LLaVA, consisting of two stages.

**Stage 1: Vision-Language Pretraining** In this stage, compression and modality pre-fusion modules are not yet applied (i.e., the $N^2$ vision tokens remain unchanged). LLaVA-Mini learns to align vision and language representations using visual caption data. The training focuses solely on the projection module while the vision encoder and LLM remain frozen (Liu et al., 2023b).

**Stage 2: Instruction Tuning** In this stage, LLaVA-Mini is trained to perform various visual tasks based on minimal vision tokens, using instruction data. Compression and modality pre-fusion are introduced to LLaVA-Mini, and all modules except the frozen vision encoder (i.e., the projection, compression, modality pre-fusion, and LLM backbone) are trained in an end-to-end manner.

## 5 EXPERIMENTS

### 5.1 EXPERIMENTAL SETTING

**Benchmarks** We evaluate LLaVA-Mini on image and video understanding tasks. Experiments are conducted on 11 image benchmarks and 7 video benchmarks. Refer to Appendix C for details.

**Baselines** LLaVA-Mini is an image/video LMM, so we compare it with several advanced image-based and video-based LMMs. Detailed description of baselines refer to Appendix D.

Table 1: Performance on 11 image-based benchmarks. 'Res.' is resolution and '#Vision Tokens' is the number of vision tokens fed to LLM backbone. '*' indicates that involving extra training data.

| Methods | LLM | Res. | #Vision Tokens | VQA$^{v2}$ | GQA | VisWiz | SciQA | VQA$^T$ | POPE | MME | MMB | SEED | LLaVA$^W$ | MM-Vet | Avg. (%) |
|---|---|---|---|---|---|---|---|---|---|---|---|---|---|---|---|
| BLIP-2 | Vicuna-13B | 224 | 32 | 65.0 | 41.0 | 19.6 | 61.0 | 42.5 | 85.3 | 1293.8 | – | 46.4 | 38.1 | 22.4 | - |
| InstructBLIP | Vicuna-7B | 224 | 32 | – | 49.2 | 34.5 | 60.5 | 50.1 | – | – | 36.0 | 53.4 | 60.9 | 26.2 | – |
| IDEFICS-9B | LLaMA-7B | 224 | 64 | 50.9 | 38.4 | 35.5 | – | 25.9 | – | – | 48.2 | – | – | – | – |
| IDEFICS-80B | LLaMA-65B | 224 | 64 | 60.0 | 45.2 | 36.0 | – | 30.9 | – | – | 54.5 | – | – | – | – |
| Qwen-VL | Qwen-7B | 448 | 256 | 78.8 | 59.3 | 35.2 | 67.1 | 63.8 | – | – | 38.2 | 56.3 | – | – | – |
| Qwen-VL-Chat | Qwen-7B | 448 | 256 | 78.2 | 57.5 | 38.9 | 68.2 | 61.5 | – | 1487.5 | 60.6 | 58.2 | – | – | - |
| SPHINX | LLaMA-13B | 224 | 289 | 78.1 | 62.6 | 39.9 | 69.3 | 51.6 | 80.7 | 1476.1 | 66.9 | 56.2 | 73.5 | 36.0 | 56.0 |
| SPHINX-2k | LLaMA-13B | 762 | 2890 | 80.7 | 63.1 | 44.9 | 70.6 | 61.2 | 87.2 | 1470.6 | 65.9 | 57.9 | 76.9 | 40.2 | 59.0 |
| mPLUG-Owl2 | LLaMA-7B | 448 | 1024 | 79.4 | 56.1 | 54.5 | 68.7 | 54.3 | - | 1450.2 | 64.5 | 57.8 | - | 36.2 | - |
| Video-LLaVA | Vicuna-7B | 224 | 256 | 74.7 | 60.3 | 48.1 | 66.4 | 51.8 | 84.4 | - | 60.9 | - | 73.1 | 32.0 | - |
| LLaVA-v1.5 | Vicuna-7B | 336 | 576 | 78.5 | 62.0 | 50.0 | 66.8 | 58.2 | 85.9 | 1510.7 | 64.3 | 58.6 | 63.4 | 30.5 | 56.3 |
| | | | | | | *LMMs with fewer vision tokens* | | | | | | | | | | |
| MQT-LLaVA | Vicuna-7B | 336 | 2 | 61.0 | 50.8 | 48.5 | 65.0 | – | 74.5 | 1144.0 | 54.4 | – | 41.7 | 19.5 | – |
| MQT-LLaVA | Vicuna-7B | 336 | 36 | 73.7 | 58.8 | 51.0 | 66.8 | – | 81.9 | 1416.3 | 63.4 | – | 59.6 | 27.8 | – |
| MQT-LLaVA | Vicuna-7B | 336 | 256 | 76.8 | 61.6 | 53.1 | 67.6 | – | 84.4 | 1434.5 | 64.3 | – | 64.6 | 29.8 | – |
| PruMerge | Vicuna-7B | 336 | 32 | 72.0 | – | – | 68.5 | 56.0 | 76.3 | 1350.3 | 60.9 | – | – | – | – |
| PruMerge++ | Vicuna-7B | 336 | 144 | 76.8 | – | – | 68.3 | 57.1 | 84.0 | 1462.4 | 64.9 | – | – | – | – |
| LLaMA-VID | Vicuna-7B | 336 | 2 | – | 55.5 | – | 68.8 | 49.0 | 83.1 | – | – | – | – | – | – |
| VoCo-LLaMA | Vicuna-7B | 336 | 1 | 72.3 | 57.0 | – | 65.4 | – | 81.4 | 1323.3 | 58.8 | 53.7 | – | – | – |
| TokenPacker | Vicuna-7B | 336 | 36 | 75.0 | 59.6 | 50.2 | – | – | 86.2 | – | 62.8 | – | – | 29.6 | – |
| | | | | | | *Ours* | | | | | | | | | | |
| LLaVA-Mini | Vicuna-7B | 336 | 1 | 77.6 | 60.9 | 56.2 | 70.4 | 57.0 | 84.4 | 1466.0 | 65.6 | 58.5 | 68.9 | 36.6 | 57.9 |
| △ *compare to LLaVA-v1.5* | | | | 0.17% | -0.9 | -1.1 | +6.1 | +3.6 | -1.3 | -1.5 | -44.7 | +1.3 | -0.1 | +5.5 | +6.1 | +1.6 |
| LLaVA-Mini-HD | Vicuna-7B | 672 | 64 | 78.9 | 61.8 | 58.5 | 69.7 | 59.1 | 85.3 | 1476.8 | 67.5 | 60.2 | 69.3 | 33.9 | 58.6 |
| △ *compare to LLaVA-v1.5* | | | | 11.1% | +0.4 | -0.2 | +8.5 | +2.9 | +0.9 | -0.6 | -33.9 | +3.2 | +1.6 | +5.9 | +3.4 | +2.4 |
| LLaVA-Mini* (Image & Video) | LLaMA-3.1-8B-Instruct | 336 | 1 | 79.0 | 61.3 | 57.4 | 83.1 | 58.5 | 85.3 | 1522.7 | 71.6 | 63.0 | 70.2 | 37.2 | 60.7 |

Table 2: Performance on video-based open-ended generative benchmarks. We report accuracy (%) for question-answer, and scores (1-5, higher is better) for question-answer and generative performance. Results marked with **bold** and underlined indicate the best and second best, respectively.

| Methods | #Frames | #Vision Tokens per Frame | Video-based Question-Answer | | | | | | Video-based Generative Performance | | | | | |
|---|---|---|---|---|---|---|---|---|---|---|---|---|---|---|
| | | | MSVD-QA | | MSRVTT-QA | | ActivityNet-QA | | Correctness | Detail | Contextual | Temporal | Consistency | Avg. |
| | | | Acc. | Score | Acc. | Score | Acc. | Score | | | | | | |
| LLaMA Adapter | 5 | 256 | 54.9 | 3.1 | 43.8 | 2.7 | 34.2 | 2.7 | 2.03 | 2.32 | 2.30 | 1.98 | 2.15 | 2.19 |
| VideoChat | 16 | 32 | 56.3 | 2.8 | 45.0 | 2.5 | 26.5 | 2.2 | 2.23 | 2.50 | 2.53 | 1.94 | 2.24 | 2.30 |
| Video-LLaMA | 16 | 64 | 51.6 | 2.5 | 29.6 | 1.8 | 12.4 | 1.1 | 1.96 | 2.18 | 2.16 | 1.82 | 1.79 | 1.99 |
| Video-ChatGPT | 100 | ∼3.6 | 64.9 | 3.3 | 49.3 | 2.8 | 35.2 | 2.7 | 2.40 | 2.52 | 2.62 | 1.98 | 2.37 | 2.37 |
| BT-Adapter | 100 | ∼2.6 | 67.5 | 3.7 | 57.0 | 3.2 | 45.7 | 3.2 | 2.68 | 2.69 | 3.27 | 2.34 | 2.46 | 2.69 |
| MovieChat | 2048 | 32 | **75.2** | 3.8 | 52.7 | 2.6 | 45.7 | 3.4 | 2.76 | 2.93 | 3.01 | 2.24 | 2.42 | 2.65 |
| LLaMA-VID | 1fps | 2 | 69.7 | 3.7 | 57.7 | 3.2 | 47.4 | 3.3 | 2.96 | **3.00** | 3.53 | 2.46 | 2.51 | 2.88 |
| Video-LLaVA | 8 | 256 | 70.7 | 3.9 | 59.2 | 3.5 | 45.3 | 3.3 | 2.87 | 2.94 | 3.44 | 2.45 | 2.51 | 2.84 |
| LLaVA-Mini | 1fps | **1** | 70.9 | **4.0** | 59.5 | **3.6** | **53.5** | **3.5** | **2.97** | 2.99 | **3.61** | **2.48** | **2.67** | **2.94** |

**Configuration** For a fair comparison, LLaVA-Mini employs the same configurations as LLaVA-v1.5 (Liu et al., 2023b), using the CLIP ViT-L/336px (Radford et al., 2021) as the vision encoder and Vicuna-v1.5-7B (Chiang et al., 2023) as the LLM backbone. The compressed hyperparameter $C$ is set to 1, meaning vision tokens are compressed to one token. The number of modality pre-fusion layers $N_{fusion}$ is set to 4. LLaVA-Mini uses the same training data as LLaVA-v1.5 (Liu et al., 2023b), using 558K caption data for pretraining and 665K instruction data for instruction tuning. The high-resolution version with 672*672 pixels (refer to Sec.4.2) is denoted as LLaVA-Mini-HD. To capture more visual details, the compressed hyperparameter $C$ of LLaVA-Mini-HD is set to 8, i.e., compressing to 64 vision tokens. For video processing, LLaVA-Mini extracts 1 frame per second (1 fps) from the video and sets $C = 1$ to represent each frame with one vision token.

To further explore the potential of LLaVA-Mini, we introduce a variant that uses the CLIP ViT-L/336px (Radford et al., 2021) as vision encoder and the advanced LLaMA-3.1-8B-Instruct (Dubey et al., 2024) as LLM backbone. During instruction tuning, we combine 665K image instruction data from LLaVA (Liu et al., 2023b), 100K video instruction data from Video-ChatGPT (Maaz et al., 2024), and part of open-source data (Li et al., 2024a), resulting in 3 million training samples. LLaVA-Mini is trained using 8 NVIDIA A800 GPUs. Training details are provided in Appendix B.

## 5.2 MAIN RESULTS

**Image-based Evaluation** We compare LLaVA-Mini with LLaVA-v1.5 across 11 benchmarks to thoroughly assess the performance of LLaVA-Mini with minimal vision tokens. The results are

Table 3: Performance on MVBench (accuracy). Detailed scores are reported in Appendix H.

| Methods | Action | Object | Position | Scene | Count | Attribute | Pose | Character | Cognition | Avg. |
|---|---|---|---|---|---|---|---|---|---|---|
| mPLUG-Owl | 28.4 | 33.0 | 25.0 | 29.0 | 29.3 | 42.0 | 24.0 | 31.0 | 25.3 | 29.7 |
| Video-ChatGPT | 32.1 | 40.7 | 21.5 | 31.0 | 28.0 | 44.0 | 29.0 | 33.0 | 30.3 | 32.7 |
| Video-LLaMA | 34.4 | 42.2 | 22.5 | 43.0 | 28.3 | 39.0 | 32.5 | 40.0 | 29.3 | 34.1 |
| VideoChat | 38.0 | 41.2 | 26.3 | 48.5 | 27.8 | 44.3 | 26.5 | 41.0 | 27.7 | 35.5 |
| LLaMA-VID | 43.4 | 36.7 | 39.8 | 22.0 | 36.5 | 37.3 | 37.5 | 34.0 | 60.5 | 41.4 |
| Video-LLaVA | 48.0 | 46.5 | 27.8 | 84.5 | 35.5 | 45.8 | 34.0 | 42.5 | 34.2 | 43.1 |
| LLaVA-Mini | 52.1 | 43.2 | 31.8 | 85.5 | 37.5 | 44.5 | 29.5 | 52.0 | 35.0 | 44.5 |

Table 4: Results on MLVU (accuracy) of long video understanding. Evaluation includes Topic Reasoning (TR), Anomaly Recognition (AR), Needle QA (NQA), Ego Reasoning (ER), Plot QA (PQA), Action Order (AO), and Action Count (AC).

Table 5: Results on EgoSchema (accuracy), a long-form video benchmark (∼ 3 minutes) for first-person view temporal reasoning.

| Methods | #Frames | Holistic | | Single Detail | | | Multi Detail | | Avg. |
|---|---|---|---|---|---|---|---|---|---|
| | | TR | AR | NQA | ER | PQA | AO | AC | |
| Avg. Video Duration (minute) | 7 | 10 | 14 | 10 | 8 | 16 | 13 | 11 |
| Max Video Duration (minute) | 20 | 543 | 139 | 20 | 13 | 137 | 130 | 143 |
| Video-ChatGPT | 100 | 26.9 | 24.0 | 40.3 | 42.0 | 29.9 | 25.1 | 31.1 | 31.3 |
| MovieChat | 2048 | 29.5 | 25.0 | 24.2 | 24.7 | 25.8 | 28.6 | 22.8 | 25.8 |
| Movie-LLM | 1fps | 30.0 | 29.0 | 29.6 | 24.7 | 24.1 | 20.5 | 24.8 | 26.1 |
| TimeChat | 96 | 23.1 | 27.0 | 24.5 | 28.4 | 25.8 | 24.7 | 32.0 | 30.9 |
| LLaMA-VID | 1fps | 50.8 | 34.5 | 30.1 | 32.7 | 32.5 | 23.9 | 27.8 | 33.2 |
| MA-LMM | 1000 | 51.9 | 35.5 | 43.1 | 38.9 | 35.8 | 25.1 | 24.3 | 36.4 |
| LLaVA-Mini | 1fps | 76.0 | 50.0 | 44.5 | 37.5 | 49.0 | 24.3 | 18.4 | 42.8 |

| Methods | #Frames | EgoSchema |
|---|---|---|
| Random | - | 20 |
| mPLUG-Owl | 16 | 31.1 |
| InternVideo | 16 | 32.1 |
| Video-ChatGPT | 100 | 36.2 |
| VideoChat | 16 | 42.2 |
| TimeChat | 96 | 33.0 |
| LLaMA-VID | 1fps | 38.5 |
| Video-LLaVA | 8 | 38.4 |
| LLaVA-Mini | 1fps | 51.2 |

reported in Table 1, where LLaVA-Mini achieves performance comparable to LLaVA-v1.5 while using only 1 vision token instead of 576. Previous efficient LMMs with fewer vision tokens often merged similar tokens directly after the vision encoder (Shang et al., 2024; Ye et al., 2024d), resulting in a significant loss of visual information and negatively impacting visual understanding of LMMs. For instance, LLaMA-VID, VoCo-LLaVA, and MQT-LLaVA, which reduce vision tokens to 1-2 tokens, lead to 5% performance drop on average. In contrast, LLaVA-Mini employs modality pre-fusion to integrate visual information into text tokens before compressing the vision tokens, achieving performance comparable to LLaVA-v1.5 at a token compression rate of 0.17%. Furthermore, LLaVA-Mini-HD shows an average performance improvement of 2.4% over LLaVA-v1.5 due to high-resolution image modeling. Note that LLaVA-Mini-HD has a computational load of 8.13 TFLOPs, which remains lower than LLaVA-v1.5's 8.55 TFLOPs. More efficiency analyses refer to Sec.5.3. Overall, LLaVA-Mini preserves strong visual understanding capabilities while compressing vision tokens, enhancing the usability of efficient LMMs in visual scenarios.

**Video-based Evaluation** We compare LLaVA-Mini with advanced video LMMs on 5 widely used video-based benchmarks. The results are reported in Table 2 and 3, where LLaVA-Mini demonstrates superior overall performance. Video LMMs such as VideoChat (Li et al., 2024c), Video-LLaVA (Lin et al., 2023a), and Video-LLaMA (Maaz et al., 2024) use much more vision tokens to represent each frame, and thereby can extract only 8-16 frames from a video due to the limited context length of LLMs, which may result in the loss of visual information in some frames. In contrast, LLaVA-Mini uses one vision token to represent each image and accordingly can extract frames from the video at a rate of 1 frame per second, thus performing better on video understanding.

**Extrapolation to Long Videos** Furthermore, we compare LLaVA-Mini with advanced long-video LMMs (can process video over 100 frames) on long-form video benchmarks, MLVU (Zhou et al., 2024b) and EgoSchema (Mangalam et al., 2023). Note that LLaVA-Mini is trained only on Video-ChatGPT instruction data and has not been exposed to any long video data, so its performance on long videos is entirely derived from the length extrapolation capabilities of its framework (Press et al., 2022). As shown in Table 4 and 5, LLaVA-Mini exhibits significant advantages in long video understanding. By representing each frame as one token, LLaVA-Mini facilitates straightforward extension to longer videos during inference. In particular, LLaVA-Mini is only trained on videos shorter than 1 minute (< 60 frames), and performs well on MLVU's long-form video, which encompasses videos over 2 hours (> 7200 frames) during inference. Overall, with one vision token per frame, LLaVA-Mini demonstrates high-quality video understanding in a more efficient manner.

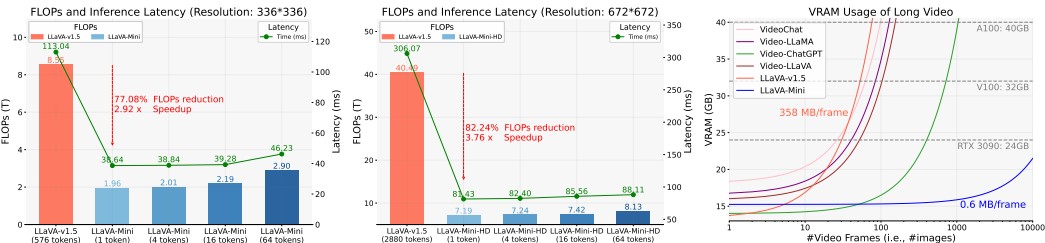

Figure 7: FLOPs and latency of LLaVA-Mini.

Figure 8: FLOPs and latency of LLaVA-Mini-HD.

Figure 9: VRAM usage (3-hour video) of LLaVA-Mini.

Table 6: Performance of LLaVA-Mini with different numbers of modality pre-fusion layers $N_{fusion}$.

| Methods | Pre-fusion #Layers | #Vision Tokens | FLOPs (T) | Performance | | |
|---|---|---|---|---|---|---|
| | | | | VQA$^{v2}$ | GQA | MMB |
| **LLaVA-v1.5** | - | 576 | 8.55 | 78.5 | 62.0 | 64.3 |
| **LLaVA-Mini (w/o pre-fusion)** | 0 | 1 | 0.96 | 72.4 | 54.2 | 57.7 |
| | 0 | 16 | 1.16 | 74.1 | 55.4 | 59.2 |
| | 0 | 64 | 1.79 | 75.3 | 56.7 | 62.1 |
| | 0 | 144 | 2.85 | 76.9 | 58.9 | 64.9 |
| **LLaVA-Mini (w/ pre-fusion)** | 1 | 1 | 1.21 | 74.8 | 55.5 | 60.4 |
| | 2 | 1 | 1.46 | 76.0 | 57.6 | 63.1 |
| | 3 | 1 | 1.81 | 76.9 | 59.1 | 64.9 |
| | 4 | 1 | 1.96 | 77.6 | 60.9 | 65.6 |

Table 7: Performance of LLaVA-Mini with various vision tokens.

| Methods | Res. | #Vision Tokens | Performance | | |
|---|---|---|---|---|---|
| | | | VQA$^{v2}$ | GQA | MMB |
| **LLaVA-v1.5** | 336 | 576 | 78.5 | 62.0 | 64.3 |
| **LLaVA-Mini** | 336 | 1 | 77.6 | 60.9 | 65.6 |
| | 336 | 4 | 77.7 | 60.9 | 66.7 |
| | 336 | 16 | 78.1 | 61.3 | 66.6 |
| | 336 | 64 | **78.5** | **61.6** | **67.5** |
| | 672 | 16 | 78.5 | 61.5 | 67.4 |
| | 672 | 64 | 78.9 | 61.8 | 67.5 |
| | 672 | 144 | 79.3 | 62.3 | 67.9 |
| | 672 | 576 | **80.0** | **62.9** | **68.1** |

## 5.3 EFFICIENCY

With the performance comparable to LLaVA-v1.5, we further explore the computational efficiency offered by LLaVA-Mini. Figures 7, 8, 9 illustrate the advantages of LLaVA-Mini in terms of computational load, inference latency, and memory usage, where FLOPs are calculated by `calflops` (Ye, 2023), and latency is tested on the A100 without any engineering acceleration techniques.

**FLOPs and Inference Latency**  As shown in Figure 7, LLaVA-Mini significantly reduces computational load by 77% compared to LLaVA-v1.5, achieving a speedup of 2.9 times. LLaVA-Mini achieves response latency lower than 40 ms, which is crucial for developing low-latency real-time LMMs. As shown in Figure 8, when modeling at high resolutions, the efficiency advantages of LLaVA-Mini are even more pronounced, yielding 82% FLOPs reduction and 3.76 times speedup.

**Memory Usage**  Memory consumption poses another challenge for LMMs, particularly in video processing. Figure 9 demonstrates the memory requirements of LMMs when processing videos of varying lengths. In previous methods, each image requires approximately 200-358 MB memory (Liu et al., 2023b; Lin et al., 2023a), limiting them to handle only about 100 frames on a 40GB GPU. In contrast, LLaVA-Mini with one vision token requires just 0.6 MB per image, enabling it to theoretically support video processing of over 10,000 frames on RTX 3090 with 24 GB of memory.

## 6 ANALYSES

### 6.1 SUPERIORITY OF MODALITY PRE-FUSION

The proposed modality pre-fusion is central to LLaVA-Mini, as it integrates visual information into text tokens in advance, facilitating extreme compression of vision tokens. To investigate the effects of modality pre-fusion, we conduct an ablation study in Table 6. Without pre-fusion, token compression results in a performance drop of around 5%, even with 144 vision tokens retained, the performance of LMMs falls short of LLaVA-v1.5. This also explains why previous token merging methods often exhibit poor performance (Ye et al., 2024d) or can only achieve a compression rate of over 40% (Shang et al., 2024). Notably, under the same FLOPs, increasing the number of pre-fusion layers yields greater benefits than increasing the number of compression vision tokens. This sup-

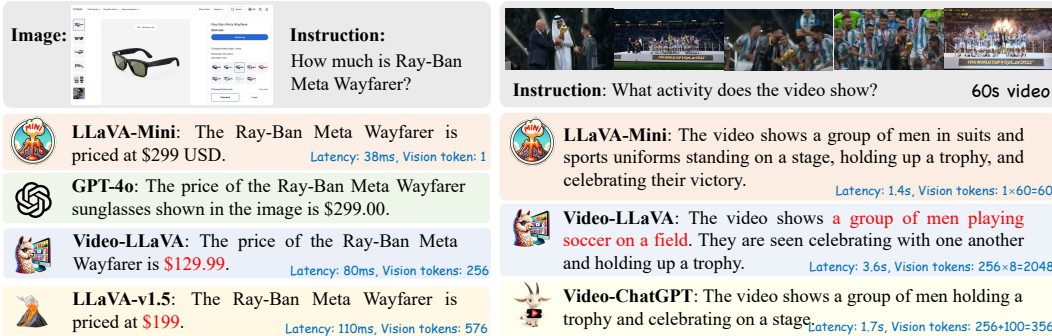

Figure 10: Case of image understanding.  Figure 11: Case of video understanding.

ports our preliminary analysis, which indicated that vision tokens exhibit varying importance across different layers and vision tokens are more critical in early layers. Investing more computational overhead in earlier stages where vision tokens are more important results in better performance.

## 6.2 Effect of Compression

LLaVA-Mini employs query-based compression to achieve a high compression ratio for vision tokens. We compare the performance of query-based compression with direct average pooling in Table 8. Query-based compression can adaptively capture important features in the image while requiring only a minimal additional computational cost, demonstrating a significant advantage. Appendix F gives a visualization of the compression process and a more detailed analysis.

Table 8: Effect of query-based compression.

| Compression | #Vision Tokens | FLOPs | Performance | | |
|---|---|---|---|---|---|
| | | | VQA$^{v2}$ | GQA | MMB |
| Average Pooling | 1 | 1.96T | 76.1 | 59.8 | 64.0 |
| Query-based | | +2.42G | **77.6** | **60.9** | **65.6** |
| Average Pooling | 4 | 2.01T | 76.9 | 60.3 | 65.1 |
| Query-based | | +2.44G | **77.7** | **60.9** | **66.7** |

## 6.3 Performance with Various Vision Tokens

LLaVA-Mini uses 1 vision token for standard images and 64 for high-resolution images. We explore the potential of LLaVA-Mini when further increasing the number of vision tokens (larger $C$) in Table 7. The results indicate that as the number of vision tokens increases, LLaVA-Mini continues to improve in performance. In particular, LLaVA-Mini outperforms LLaVA-v1.5 when both using 576 vision tokens, demonstrating its effectiveness when computational resources are plentiful.

## 6.4 Case Study

Figures 10 and 11 present examples of LLaVA-Mini in image and video understanding tasks (refer to Appendix G for more cases). Despite using only one vision token, LLaVA-Mini performs effectively in capturing visual details, such as accurately identifying price information (OCR) in website screenshots. For video understanding, Video-LLaVA extracts 8 frames per video, regardless of video duration (Lin et al., 2023a). Training on only 8 frames (sometimes missing key frames) can cause hallucinations (Khattak et al., 2024), encouraging LMM to forge information beyond the extracted frames. For instance, given a celebration scene, Video-LLaVA mistakenly imagines "*a group of men playing soccer on a field*" before the celebration. This fixed-length frame extraction is a forced compromise due to the large number of vision tokens required per image while LLM's context length is limited. In contrast, LLaVA-Mini, utilizing just one vision token per frame, can process videos at 1 fps, resulting in more robust video understanding. Overall, LLaVA-Mini ensures strong visual understanding while enhancing efficiency, making it a practical solution for multimodal interaction.

## 7 Conclusion

We introduce LLaVA-Mini, an efficient LMM with minimal vision tokens. LLaVA-Mini excels in image and video understanding while exhibiting superiority in computational efficiency, inference latency, and memory usage, facilitating the real-time multimodal interaction with efficient LMMs.

## ACKNOWLEDGMENTS

We thank all the anonymous reviewers for their insightful and valuable comments. This work was supported by National Natural Science Foundation of China (Grant No.62376260).

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

## A    DETAILED SETTING OF PRELIMINARY ANALYSES

In Sec.3.2, we analyze the importance of visual tokens in LMMs from an attention-based perspective to inform strategies for compressing vision tokens. Here, we give a detailed introduction of the experimental setup for the attention analysis.

We focus on the LLaVA series architecture, where the input tokens to the LLM are composed of instruction tokens, vision tokens, and response tokens, as shown in Eq.(1). We compute the average attention received by each type of token to reveal how the importance of different token categories changes across layers.

**Calculation of Attention Weights**    Formally, we denote the attention of the $i^{th}$ token $h_i$ to the $j^{th}$ token $h_j$ as $a_{ij}$, where $a_{ij}$ is the average attention across all attention heads. All tokens fed to the LLM are divided into instruction tokens, vision tokens, and response tokens according to inputs type, denoted as sets $T_{instruction}$, $T_{vision}$, and $T_{response}$ respectively. Finally, denoted the target and source token types as $tgt\_type, src\_type \in \{$instruction, vision, response$\}$, the average attention weights from $tgt\_type$ type tokens to $src\_type$ type tokens in our analyses are calculated as:

$$\text{Attn}(tgt\_type \to src\_type) = \frac{\sum_{h_i \in T_{tgt\_type}} \sum_{h_j \in T_{src\_type}} a_{ij}}{\sum_{h_i \in T_{tgt\_type}} \mathbb{1}_{\sum_{h_j \in T_{src\_type}} a_{ij} > 0}}, \tag{4}$$

$$\text{where } \mathbb{1}_{\sum_{h_j \in T_{src\_type}} a_{ij} > 0} = \begin{cases} 1 & \text{if } \sum_{h_j \in T_{src\_type}} a_{ij} > 0 \\ 0 & \text{otherwise} \end{cases} \tag{5}$$

Specifically, $\sum_{h_i \in T_{tgt\_type}} \sum_{h_j \in T_{src\_type}} a_{ij}$ calculates the sum of attention weights from all $tgt\_type$ type tokens to all $src\_type$ type tokens, $\sum_{h_i \in T_{tgt\_type}} \mathbb{1}_{\sum_{h_j \in T_{src\_type}} a_{ij} > 0}$ counts the number of $tgt\_type$ type tokens, thus $\text{Attn}(tgt\_type \to src\_type)$ represents the average attention weight from $tgt\_type$ type tokens to $src\_type$ type tokens. $\text{Attn}(tgt\_type \to src\_type)$ is consistent with the legend in Figure 2.

**Calculation of Attention Entropy**    The calculation of attention entropy is similar to that of attention weights, with the key difference being the addition of a normalization step. When computing the entropy of a specific type of token (e.g., vision tokens), the sum of attention weights for this token type may not equal 1. Thus, we perform a normalization on the attention of these tokens (e.g., vision tokens) to ensure the definition of entropy is satisfied.

In practice, for LLaVA-v1.5 (pad) (Liu et al., 2023b) and LLaVA-NeXT (anyres) (Liu et al., 2024b), which may involve different resolution vision inputs, we use their original settings. In our analysis, we do not further distinguish between different types of vision tokens (e.g., global or local), but treat them collectively as vision tokens.

## B    TRAINING DETAILS

**Implementation Details**    The compression method of LLaVA-Mini can be easily plugged into existing multi-modal pipelines, as it only requires the addition of two extra modules (the compression module and the modality pre-fusion module) before the LLM, while the other components (such as the vision encoder, the LLM, and the training loss) remain unchanged. The pre-fusion module applies the same decoder-only architecture as the LLM, including both the structure and hyper-parameters. The motivation behind this setting is to ensure flexible adaptation to existing LLM frameworks and other acceleration techniques.

**Training**    The overall training process follows a two-stage paradigm similar to LLaVA, consisting of vision-language pretraining followed by instruction tuning. Table 9 reports the two-stage training details of LLaVA-Mini.

## C    BENCHMARKS

We conduct a comprehensive evaluation of LLaVA-Mini, including both image and video understanding benchmarks.

Table 9: Training details of LLaVA-Mini.

| Settings | | Stage1 Vision-Language Pretraining | Stage2 Instruction Turning |
|---|---|---|---|
| **Modules** | Vision Encoder | Frozen | Frozen |
| | Projection | Trainable | Trainable |
| | Large Language Model | Frozen | Trainable |
| | Compression | N/A | Trainable |
| | Modality Pre-fusion | N/A | Trainable |
| **Hyperparameters** | Batch Size | 256 | 256 |
| | Learning Rate | - | 1e-4 |
| | MM Learning Rate | 1e-3 | 1e-5 |
| | Schedule | Cosine decay | |
| | Warmup Ratio | 0.03 | |
| | Optimizer | AdamW | |
| | Epoch | 1 | 2 |

## C.1 IMAGE-BASED BENCHMARKS

Following the LLaVA framework (Liu et al., 2023b), we conduct experiments on 11 widely adopted benchmarks, including VQA-v2 ($\text{VQA}^{\text{v2}}$) (Goyal et al., 2017), GQA (Hudson & Manning, 2019), VisWiz (Gurari et al., 2018), ScienceQA-IMG (SciQA) (Lu et al., 2022), TextVQA ($\text{VQA}^{\text{T}}$) (Singh et al., 2019), POPE (Li et al., 2023c), MME (Fu et al., 2024), MMBench (MMB) (Liu et al., 2024c), SEED-Bench (SEED) (Li et al., 2024b), LLaVA-Bench-in-the-Wild ($\text{LLaVA}^{\text{W}}$) (Liu et al., 2023a), and MM-Vet (Yu et al., 2023), which cover a diverse range of visual tasks. The evaluation pipelines for all benchmarks are consistent with those used in LLaVA.

## C.2 VIDEO-BASED BENCHMARKS

**Video-based Generative Performance Benchmark** For video-based evaluation, we conduct experiments on video open-ended question-answering benchmarks, including MSVD-QA (Chen & Dolan, 2011), MSRVTT-QA (Xu et al., 2016), and ActivityNet-QA (Caba Heilbron et al., 2015). Furthermore, we use the video-based generative performance benchmark (Maaz et al., 2024) to assess the performance of LLaVA-Mini across five dimensions: correctness, detail orientation, contextual understanding, temporal understanding, and consistency. The evaluation pipelines for both the open-ended question-answering and the generative performance benchmarks adhere to Video-ChatGPT (Maaz et al., 2024), employing the GPT model (gpt-3.5-turbo version) to evaluate the accuracy of responses (True or False) and to assign a score ranging from 1 to 5 for response, where higher scores indicate superior performance.

**MVBench** (Li et al., 2024d) MVBench is a comprehensive benchmark for multimodal video understanding that encompasses 20 challenging tasks. The evaluation aspects of MVBench include Action (such as Action Sequence, Action Prediction, Action Antonym, Fine-grained Action, and Unexpected Action), Object (Object Existence, Object Interaction, Object Shuffle), Position (Moving Direction, Action Localization), Scene (Scene Transition), Count (Action Count, Moving Count), Attribute (Moving Attribute, State Change), Pose (Fine-grained Pose), Character (Character Order), and Cognition (Egocentric Navigation, Episodic Reasoning, Counterfactual Inference). The evaluation of MVBench employs a multiple-choice format, using accuracy as the metric.

**MLVU** (Zhou et al., 2024b) MLVU is a comprehensive benchmark for multi-task long video understanding. The evaluation aspects of MLVU include Topic Reasoning (TR), Anomaly Recognition (AR), Needle QA (NQA), Ego Reasoning (ER), Plot QA (PQA), Action Order (AO), and Action Count (AC). The evaluation of MLVU also employs a multiple-choice format, using accuracy as the metric.

**EgoSchema** (Mangalam et al., 2023) EgoSchema is a long-form video question-answering dataset, which serves as a benchmark for assessing the long video understanding capabilities of first-person videos. The evaluation of EgoSchema also employs a multiple-choice format, using accuracy as the metric.

## D    INTRODUCTION TO BASELINES

LLaVA-Mini is an image and video LMM, so we compare it with several advanced image-based and video-based LMMs.

### D.1    IMAGE-BASED LMMs

We compare LLaVA-Mini with LLaVA-v1.5 (Liu et al., 2023b) and other advanced LMMs of similar data and model scales, including BLIP-2 (Li et al., 2023a), InstructBLIP (Liu et al., 2024a), IDEFICS (Laurençon et al., 2023), Qwen-VL (Bai et al., 2023), Qwen-VL-Chat (Bai et al., 2023), SPHINX (Lin et al., 2023b), mPLUG-Owl2 (Ye et al., 2024c).

**LMMs with Fewer Vision Tokens**    Additionally, we assess LLaVA-Mini against various efficient LMMs that utilize fewer vision tokens, showing advantages in compression rate and performance. Most of these models share the same architecture and training data as LLaVA, primarily focusing on the merging of vision tokens in the vision encoder. These efficient LMMs are introduced as follows.

**MQT-LLaVA** (Hu et al., 2024) introduces a flexible query transformer that allows encoding an image into a variable number of visual tokens (up to a predefined maximum) to adapt to different tasks and computational resources.

**PruMerge** (Shang et al., 2024) reduces visual tokens in LMMs by identifying and merging important tokens based on the attention sparsity in vision encoder. PruMerge has a variant, named PruMerge++, which enhances the original PruMerge method by evenly adding more vision tokens (about 144 vision tokens) to further improve performance.

**LLaMA-VID** (Li et al., 2023b) LLaMA-VID compresses the instruction and image into one token respectively, with a total of two tokens representing each image, thus facilitating the understanding of longer videos.

**VoCo-LLaMA** (Ye et al., 2024d) compresses all vision tokens using language models, significantly improving computational efficiency.

**TokenPacker** (Li et al., 2024e) is a visual projector that efficiently reduces visual tokens by 80% using a coarse-to-fine approach.

Previous methods have often focused on reducing the number of vision tokens output by the vision encoder. LLaVA-Mini takes this a step further by shifting attention to how vision tokens and text tokens interact within the LLM backbone. Based on this insight, we propose modality pre-fusion, which enables better performance even under the extreme compression of reducing vision tokens to just one token.

### D.2    VIDEO-BASED LMMs

LLaVA-Mini can also perform high-quality video understanding, so we compare LLaVA-Mini with the current advanced video LMMs, including LLaMA-Adaptor (Zhang et al., 2024), InternVideo (Wang et al., 2022), VideoChat (Li et al., 2024c), Video-LLaMA (Zhang et al., 2023), mPLUG-Owl (Ye et al., 2024b), Video-ChatGPT (Maaz et al., 2024), BT-Adapor (Liu et al., 2023c), LLaMA-VID (Li et al., 2023b), and Video-LLaVA (Lin et al., 2023a).

We also compare LLaVA-Mini with several video LMMs specifically designed for long videos, including MovieChat (Song et al., 2024a), Movie-LLM (Song et al., 2024b), TimeChat (Ren et al., 2023), MA-LMM (He et al., 2024). Note that among these video LMMs, LLaVA-Mini and Video-LLaVA can complete image and video understanding with a unified model.

## E    EXTENDED EXPERIMENTAL RESULTS

### E.1    EFFECT OF COMPRESSION MODULE

To verify the effectiveness of the compression module, we compared the compression module in LLaVA-Mini with previous advanced token merging methods. To ensure a fair comparison of

Table 10: Comparison of LLaVA-Mini with previous token merging methods.

| Methods | #Vision Tokens | Performance | | |
|---------|---------------|-------|-----|-----|
| | | VQA$^{v2}$ | GQA | MMB |
| **MQT-LLaVA** | 2 | 61.0 | 50.8 | 54.4 |
| **MQT-LLaVA** | 36 | 73.7 | 58.8 | 63.4 |
| **MQT-LLaVA** | 256 | 76.8 | 61.6 | 64.3 |
| **PruMerge** | 32 | 72.0 | - | 60.9 |
| **PruMerge++** | 144 | 76.8 | - | 64.9 |
| **LLaVA-Mini** | 1 | 72.4 | 54.2 | 57.7 |
| **LLaVA-Mini** | 16 | 74.1 | 55.4 | 59.2 |
| **LLaVA-Mini** | 64 | 75.3 | 56.7 | 62.1 |
| **LLaVA-Mini** | 144 | 76.9 | 58.9 | 64.9 |

token compression performance, we remove the modality pre-fusion module from LLaVA-Mini for the comparison with SOTA token merging methods, including PruMerge (Shang et al., 2024), PruMerge++ (Shang et al., 2024), and MQT-LLaVA (Hu et al., 2024). Specifically, PruMerge applies the widely-used token merge (ToMe) technique (Bolya et al., 2023) on ViT, PruMerge++ improves upon PruMerge by uniformly sampling additional vision tokens, and MQT-LLaVA employs Matryoshka representation learning to compress vision tokens.

As shown in Table 10, LLaVA-Mini's compression module outperforms PruMerge, PruMerge++, and MQT-LLaVA at the same compression rate, showing the advantages of query-based compression.

### E.2 EFFECT OF MODALITY PRE-FUSION

Table 11: Performance of LLaVA-Mini when using only pre-fusion module without compression.

| Methods | #Vision Tokens | Performance | | |
|---------|---------------|-------|-----|-----|
| | | VQA$^{v2}$ | GQA | MMB |
| **LLaVA-v1.5** | 576 | 78.5 | 62.0 | 64.3 |
| **LLaVA-Mini (w/o compression)** | 576 | **80.0** | **62.9** | **66.2** |

To validate the effect of the pre-fusion module, we remove the compression module and retained only the modality pre-fusion module, thereby comparing with LLaVA-v1.5 while both using 576 vision tokens. As shown in Table, when using only the pre-fusion module without compression, LLaVA-Mini achieves superior performance compared to LLaVA-v1.5 with both using 576 vision tokens, demonstrating the effectiveness of the pre-fusion module.

### E.3 WHY PREFORMING COMPRESSION AND PRE-FUSION OUTSIDE LLM BACKBONE?

LLaVA-Mini performs compression and modality pre-fusion before the LLM backbone. The motivation for conducting these processes outside the LLM backbone, rather than conducting at the $L^{th}$ layer within the LLM, stems from two key considerations:

- Vision representations after the $L^{th}$ layers contain contextual information, which hinders the compression module: After the vision tokens are fed into the LLM, the early layers cause the visual representations to carry contextual information. Applying query-based compression on top of these representations makes it difficult for the compression module to distinguish between different vision tokens.

- The inter-layer operations within the LLM may not be compatible with existing acceleration frameworks: One of the main motivations for placing the compression and pre-fusion modules outside the LLM backbone in LLaVA-Mini is to keep the LLM backbone unchanged. This design allows for compatibility with nearly all existing LLM acceleration technologies and frameworks, further enhancing efficiency.

Table 12: Comparison of performing compression and pre-fusion outside or within LLM backbone.

| Methods | #Vision Tokens | FLOPs (T) | Performance | | |
|---|---|---|---|---|---|
| | | | VQA$^{v2}$ | GQA | MMB |
| **LLaVA-Mini** | 1 | 1.96 | **77.6** | **60.9** | **65.6** |
| **LLaVA-Mini (perform compression and pre-fusion within LLM)** | 1 | 1.84 | 76.3 | 60.1 | 64.5 |

We also conduct a comparison between LLaVA-Mini and LLaVA-Mini (compression and pre-fusion within LLM) in Table 12. The results demonstrate that the configuration of LLaVA-Mini is more advantageous. We will incorporate this result and the architectural motivation into the manuscript as per your recommendation.

## E.4 Efficiency across Various Hardware

Table 13: Inference latency (millisecond) of LLaVA-Mini on various hardware platforms.

| Methods | #Vision Tokens | RTX 3090 (24G) | A100 (40G) | A800 (80G) |
|---|---|---|---|---|
| **LLaVA-v1.5** | 576 | 198.75 | 113.04 | 87.43 |
| **LLaVA-Mini** | 1 | 64.52 | 38.64 | 27.43 |
| | 4 | 65.52 | 38.84 | 27.71 |
| | 16 | 68.97 | 39.28 | 28.92 |
| | 64 | 80.10 | 46.23 | 34.65 |

The efficiency improvements brought by LLaVA-Mini stem from reduced computational load (FLOPs), which is consistent across different hardware platforms. To demonstrate the scalability of model efficiency across different hardware platforms, we compute the inference latency of LLaVA-Mini on three hardware platforms: RTX 3090, A100, and A800. As shown in Table 13, the efficiency improvements brought by LLaVA-Mini are scalable across these hardware platforms.

## E.5 Computational Overhead of Each Component

Table 14: Computational overhead (FLOPs) of each component in LLaVA-Mini.

| Methods | Res. | FLOPs (T) | | | | | |
|---|---|---|---|---|---|---|---|
| | | Vision Encoder | Projection | Compression | Pre-fusion | LLM | Total |
| **LLaVA-v1.5** | 336 | 0.349 | 0.024 | - | - | 8.177 | 8.55 |
| **LLaVA-Mini** | 336 | 0.349 | 0.024 | 0.001 | 0.125 | 1.460 | 1.96 |
| **LLaVA-v1.5** | 672 | 1.745 | 0.121 | - | - | 38.623 | 40.49 |
| **LLaVA-Mini** | 672 | 1.745 | 0.121 | 0.009 | 1.183 | 4.131 | 7.19 |

LLaVA-Mini significantly reduces the computational load of LMMs by decreasing the number of vision tokens. To further study the proportion of computational load contributed by each component in LLaVA-Mini, we compute the FLOPs of each module, as shown in Table 14. The proposed compression module and pre-fusion module incur minimal computational cost, while the computation required by the LLM backbone is significantly reduced.

## F Visualization of Compression

LLaVA-Mini introduces query-based compression to adaptively compress vision tokens while preserving essential information. The learnable queries in compression module interact with all vision tokens through cross-attention to capture key visual information. To verify the effectiveness of the proposed compression, Figure 12 visualizes the cross-attention during the compression process. Across various image types and styles (e.g., photographs, text, screenshots, and cartoons),

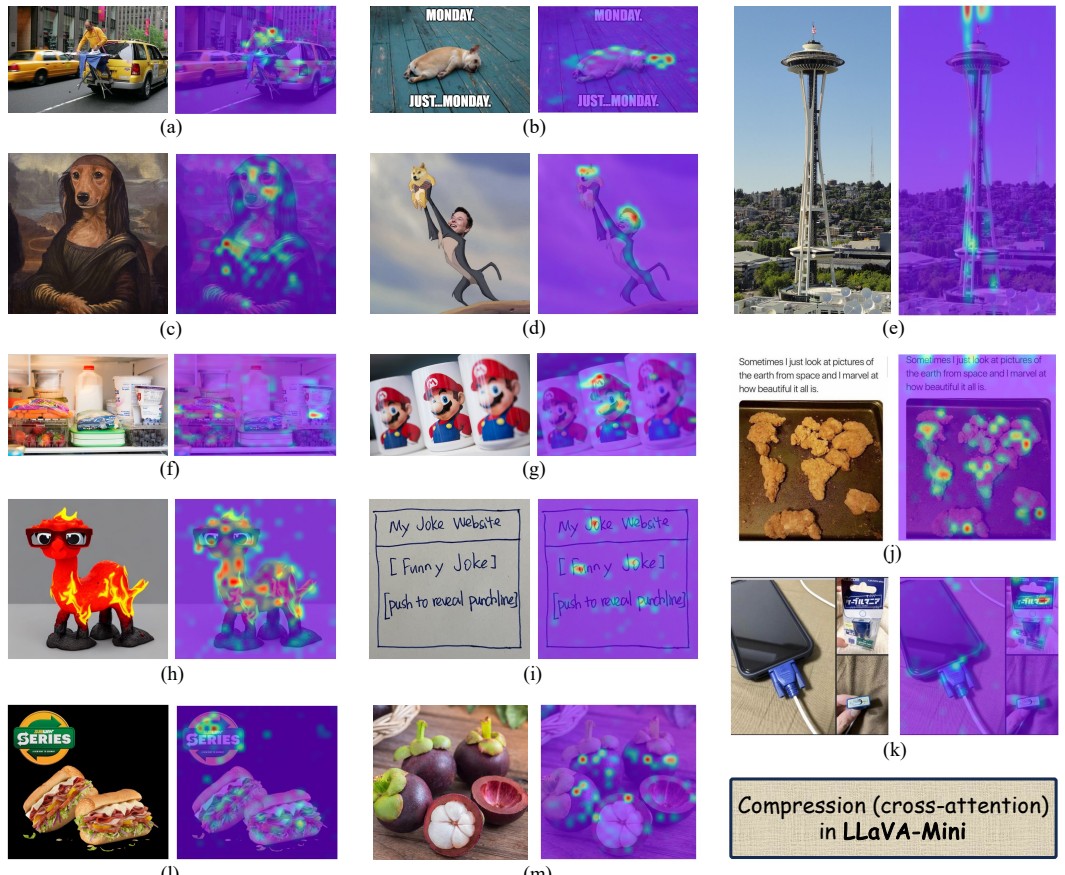

Figure 12: Visualization of the cross-attention in the compression module introduced in LLaVA-Mini. The left side is the original image, and the right side is the cross-attention distribution heat map, where brighter areas are more heavily weighted during compression. The example images are all from the LLaVA-Bench-in-the-Wild benchmark.

LLaVA-Mini's compression exhibits strong interpretability, effectively extracting key visual information from images. In cases where critical information is concentrated (such as (b), (d), (h), (i) in Figure 12), LLaVA-Mini focuses on these key locations. Conversely, in cases where the main object is unclear. (such as (f), (j), (i), (m) in Figure 12), LLaVA-Mini exhibits a more dispersed attention pattern during the compression process, thereby preserving a broader range of visual information.

In particular, for complex image like Figure 12(k), which contain multiple sub-figures with logical relationships, the proposed compression module adaptively pay attention to the VGA-shaped charger, the product name on the charger packaging, and the charging port of the charger, demonstrating the superiority of the proposed compression. Overall, compared to compression based on average pooling, query-based compression allows LLaVA-Mini to adaptively assign greater weight to key information, effectively retaining important visual details after compression.

## G  MORE CASES

**Image Understanding**  Figure 13 illustrates an example of LLaVA-Mini's capabilities in more complex image reasoning. The image in Figure 13 incorporates features such as metaphor and counterfactual reasoning, requiring LMMs to accurately interpret the visual information and reason about the humorous aspects based on the entities present in the scene. The results demonstrate that only LLaVA-Mini and GPT-4o successfully capture the phrases "*the musicians are performing on a giant violin*" and "*The violin, being a large and buoyant object, is placed on the water*", while both Video-LLaVA and LLaVA-v1.5 fail to understand this image. In terms of the perception of

**Image:** 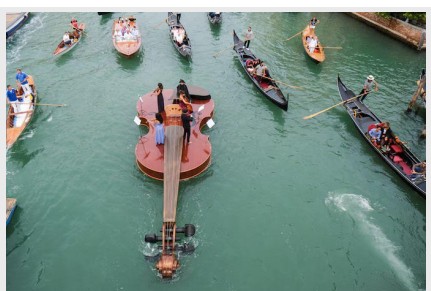

**Instruction:**

What is funny about this image? Describe it panel by panel.

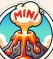 **LLaVA-Mini**: The image is a creative and humorous take on a traditional concert setting, where the musicians are performing on a giant violin, which is a whimsical and unexpected sight. The violin, being a large and buoyant object, is placed on the water, and the musicians are seated on it as if it were a stage. The presence of the gondolas and the water setting adds to the humor, as it's not a typical environment for a concert. The musicians' serious expressions and the formal attire of the performers contrast with the absurdity of the situation, which is likely intended to be a playful and entertaining spectacle. The image plays on the concept of a concert in a unique and imaginative way, using the visual pun of the violin's shape and the water setting to create a humorous and memorable scene.

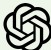 **GPT-4o**: This image shows a humorous and creative scene of a boat shaped like a giant violin floating on a canal, surrounded by traditional gondolas. Here's a panel-by-panel description:

Central Focus: The Violin Boat – The main object of humor is the boat designed to look like a massive violin or stringed instrument. On top of it, there are people standing or possibly playing music. This design turns an ordinary boat into a whimsical, oversized musical instrument.

Surrounding Gondolas – Around the violin-shaped boat, regular gondolas can be seen with people rowing and sitting in them. These gondolas look tiny compared to the giant violin boat, adding to the comedic contrast. The gondoliers and passengers in these smaller boats seem to be watching or escorting the violin boat, enhancing the spectacle.

Water and Context – The canal is typical of places like Venice, known for its waterways and gondolas. The idea of transforming a boat into a musical instrument on such a canal adds to the absurdity and charm of the scene.

The humor comes from the unusual sight of a giant floating violin on a canal, combining the elegance of music with a quirky, oversized version of an everyday object in a whimsical, real-world setting.

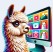 **Video-LLaVA**: The image is a panel of four pictures that showcase a humorous scene of a boat with a man playing a musical instrument on it. The boat is floating in a body of water, and the man is playing a guitar. The funny aspect of the scene is that the boat is shaped like a guitar, making it a unique and amusing sight. The four pictures capture different moments of the man playing the guitar on the boat, showcasing the creativity and humor of the scene.

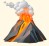 **LLaVA-v1.5**: The image is a panel of a comic strip featuring a man in a boat, paddling with a paddle. The boat is surrounded by other boats, and the man is wearing a tie. The funny aspect of this image is that the man is wearing a tie while paddling, which is an unusual combination of attire for a boat ride. The scene is set in a river, and there are several people in the boats around him. The man's attire and the presence of multiple boats create a humorous and unexpected visual.

Figure 13: Example demonstrating LLaVA-Mini's image understanding capability on more complex image reasoning. Output marked in red indicates factual errors.

entities in the picture, both Video-LLaVA and LLaVA-v1.5 exhibit hallucinations in their descriptions. Specifically, Video-LLaVA erroneously interprets the image as "*The image is a panel of four pictures*" and "*the man is playing a guitar*", while LLaVA-v1.5 fails to recognize the presence of the violin entirely.

Figure 14 illustrates an example of LLaVA-Mini's capabilities in a more challenging OCR task. The text in the image is presented in an unusual cursive handwriting style, which can significantly hinder the recognition quality of LMMs. For this challenging OCR case, both LLaVA-Mini and GPT-

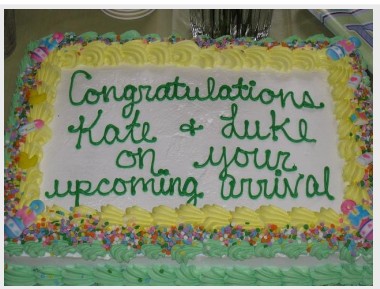

**Image:**

**Instruction:**

What's the text on the cake?

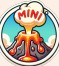 **LLaVA-Mini**: Congratulations Kate & Luke on your upcoming arrival

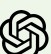 **GPT-4o**: The text on the cake says: "Congratulations Kate & Luke on your upcoming arrival"

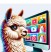 **Video-LLaVA**: The text on the cake reads "Congratulations Kate & Duke on your upcoming arrival!"

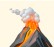 **LLaVA-v1.5**: The text on the cake reads "Congratulations Kate & Luke on your upcoming wedding!"

Figure 14: Example demonstrating LLaVA-Mini's image understanding capability on more challenging OCR task. Output marked in red indicates recognition errors.

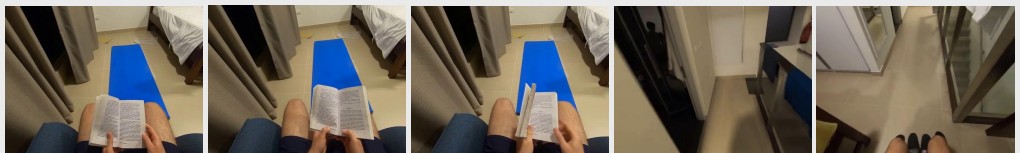

**Instruction**: Describe what happen in this first person video?

**3min video**

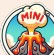 **LLaVA-Mini**: The first-person video captures a moment of solitude and relaxation, likely in a private space such as a bedroom or a hotel room. The individual is seated comfortably on a chair, engrossed in reading a book, which suggests a moment of leisure or study. The presence of the yoga mat indicates that the person may have been or plans to engage in physical activity, possibly as part of a routine or for relaxation. The room's minimalistic decor and the person's casual attire suggest a setting that is informal and personal. The video does not provide any explicit narrative or context beyond these observations, so any further interpretation would be speculative.

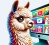 **Video-LLaVA**: In this video, a man is seen sitting on a chair and reading a book. He then stands up and walks towards the bathroom where he takes a shower. After that, he is seen sitting on the floor and reading the book again. The video seems to be a simple yet peaceful depiction of a man's daily routine. The man appears to be calm and relaxed throughout the video, and the setting is minimalistic, with no other characters or objects present. The video's focus is solely on the man and his actions, and the viewer is left to interpret the meaning behind the scenes. Overall, the video is a simple yet intriguing portrayal of a man's daily routine, and it is sure to leave the viewer with a sense of calm and relaxation.

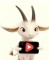 **Video-ChatGPT**: In the video, a man is seen sitting on a chair and reading a book. He then puts the book down and stands up. The video then shows him walking down a hallway and entering a room.

Figure 15: Example demonstrating LLaVA-Mini's video understanding capability on first-person view video. Output marked in red indicates factual errors.

4o accurately identify the text in the image, particularly with LLaVA-Mini using only one vision token. In contrast, Video-LLaVA and LLaVA-v1.5 incorrectly recognize "*Duke*" and "*wedding*", and erroneously add an exclamation mark "!" at the end. Overall, LLaVA-Mini demonstrates superior performance in perceiving and reasoning about visual information.

**Video Understanding**    Figure 15 illustrates an example of LLaVA-Mini's capabilities in processing longer first-person video. The results show that LLaVA-Mini exhibits a more comprehensive and detailed understanding of the video, effectively capturing entities in the room, such as the yoga mat. In contrast, Video-LLaVA mistakenly imagines "*he takes a shower*" due to its limitation of extracting only 8 frames from the video. Video-ChatGPT provides much shorter responses, lacking some detailed information. Overall, LLaVA-Mini exhibits a superior understanding of the video.

# H    DETAILED RESULTS ON MVBENCH

Table 15 reports the detailed results on each subset of MVBench, corresponding to Table 3.

Table 15: Detailed results on 20 subsets of MVBench.

| Spatial | Temporal | mPLUG-Owl | Video-ChatGPT | Video-LLaMA | VideoChat | LLaMA-VID | Video-LLaVA | LLaVA-Mini |
|---|---|---|---|---|---|---|---|---|
| **Average** | | 29.7 | 32.7 | 34.1 | 35.5 | 41.4 | 43.1 | 44.5 |
| **Action** | Action Sequence | 22.0 | 23.5 | 27.5 | 33.5 | 63.5 | 44.5 | 44.5 |
| | Action Prediction | 28.0 | 26.0 | 25.5 | 26.5 | 42.0 | 50.0 | 44.5 |
| | Action Antonym | 34.0 | 62.0 | 51.0 | 56.0 | 26.5 | 49.0 | 76.0 |
| | Fine-grained Action | 29.0 | 22.5 | 29.0 | 33.5 | 43.0 | 42.0 | 37.0 |
| | Unexpected Action | 29.0 | 26.5 | 39.0 | 40.5 | 42.0 | 54.5 | 58.5 |
| **Object** | Object Existence | 40.5 | 54.0 | 48.0 | 53.0 | 39.0 | 52.5 | 50.0 |
| | Object Interaction | 27.0 | 28.0 | 40.5 | 40.5 | 34.5 | 46.5 | 50.0 |
| | Object Shuffle | 31.5 | 40.0 | 38.0 | 30.0 | 36.5 | 40.5 | 29.5 |
| **Position** | Moving Direction | 27.0 | 23.0 | 22.5 | 25.5 | 44.0 | 27.0 | 31.0 |
| | Action Localization | 23.0 | 20.0 | 22.5 | 27.0 | 35.5 | 28.5 | 32.5 |
| **Scene** | Scene Transition | 29.0 | 31.0 | 43.0 | 48.5 | 22.0 | 84.5 | 85.5 |
| **Count** | Action Count | 31.5 | 30.5 | 34.0 | 35.0 | 44.5 | 44.5 | 35.0 |
| | Moving Count | 27.0 | 25.5 | 22.5 | 20.5 | 28.5 | 26.5 | 40.0 |
| **Attribute** | Moving Attribute | 40.0 | 39.5 | 32.5 | 42.5 | 19.0 | 53.0 | 48.0 |
| **Pose** | State Change | 44.0 | 48.5 | 45.5 | 46.0 | 55.6 | 38.5 | 41.0 |
| | Fine-grained Pose | 24.0 | 29.0 | 32.5 | 26.5 | 37.5 | 34.0 | 29.5 |
| **Character** | Character Order | 31.0 | 33.0 | 40.0 | 41.0 | 34.0 | 42.5 | 52.0 |
| **Cognition** | Egocentric Navigation | 26.0 | 29.5 | 30.0 | 23.5 | 84.5 | 32.5 | 31.0 |
| | Episodic Reasoning | 20.5 | 26.0 | 21.0 | 23.5 | 40.5 | 38.0 | 38.0 |
| | Counterfactual Inference | 29.5 | 35.5 | 37.0 | 36.0 | 56.5 | 32.0 | 36.0 |

