# OpenReview forum: "LLaVA-Mini: Efficient Image and Video Large Multimodal Models with One Vision Token"
_ICLR.cc/2025/Conference — ICLR 2025 Poster_

### Official Review · Reviewer_CwER · 2024-10-19

**Soundness:** 3
**Presentation:** 4
**Contribution:** 4
**Rating:** 8
**Confidence:** 5

**Summary:**

This paper proposes an efficient LMM with minimal vision tokens. 1/ To achieve a high compression ratio of vision tokens while preserving visual information, the authors first analyze how LMMs understand vision tokens and find that most vision tokens only play a crucial role in the early layers, where they fuse visual information into text tokens. 2/ LLaVA-Mini introduces a novel modality pre-fusion to fuse visual information into text tokens in advance before feeding into LLM and a compression module to reduce #vision tokens into minimal ones. 3/ Experiments across 11 image-based and 7 video-based benchmarks demonstrate that LLaVA-Mini outperforms LLaVA-v1.5 with just 1 vision token instead of 576. Efficiency analyses reveal that LLaVA-Mini can reduce FLOPs by 77%, deliver low-latency responses within 40 milliseconds, and process over 10,000 frames of video on GPU hardware with 24GB of memory.

**Strengths:**

1/ The problem of how to improve efficiency of MLLM is important.

2/ The authors conducted a very interesting and detailed analysis of how LMMs understand vision tokens and find that most vision tokens only play a crucial role in the early layers, where they fuse visual information into text tokens.

3/ LLaVA-Mini introduces a modality pre-fusion module to fuse visual information into text tokens in advance before feeding into LLM and a compression module to reduce #vision tokens into minimal ones.

4/ Experimental results are strong and convincing.

**Weaknesses:**

1/ According to Sec 3, if I understand correctly, for each downstream task, this paper trains a new, independent compression and modality pre-fusion modules. In other words, for another task, we need to train another model. I wonder, how does this approach differ from task-specific distillation? It seems the reason why the efficiency can be improved significantly is because of removing unnecessary tokens for the target downstream task. Instead, would it be possible to train generic compression and modality pre-fusion modules in stage 1 that can be generalised to various downstream tasks?

2/ For the proposed approach, a new modality pre-fusion module has been introduced. I wonder if this is necessary. Instead, can we re-use the first few layers, which have high activation for vision tokens, as the fusion module. To further include the compression module, in the first few layers of LLM, the part digesting vision tokens can have two pathways, one pathway is to output the compressed tokens while the other pathway is to be gradually fused together with the text tokens.

3/ When it comes to video, there are some recent streaming video LLM works such as "VideoLLM-online: Online Video Large Language Model for Streaming Video. CVPR 2024" which also advocates the idea of representing each frame with minimal vision tokens to improve efficiency. It might be worthwhile to compare or just discuss.

**Questions:**

Kindly refer to the weakness sec above

---

> ### Author Response · Authors · 2024-11-20
> **Response to Reviewer CwER (Part 1 of 2)**
>
> Thanks for your careful and valuable comments, and we will refine the paper following your suggestions. Following, we will respond to your questions in detail.
>
> &nbsp;
>
> **Q1**: For each downstream task, this paper trains a new, independent compression and modality pre-fusion modules? Would it be possible to train generic compression and modality pre-fusion modules in stage 1 that can be generalised to various downstream tasks?
>
> **A1**: We apologize for any confusion. In fact, **LLaVA-Mini handles all visual tasks with a single model**, including vision projection, the compression module, the modality pre-fusion module, and the LLM backbone. In LLaVA-Mini, we do not train separate compression and modality pre-fusion modules for different downstream visual tasks.
>
> The misunderstanding may have arisen from our description of the two-stage training process. Here, I give further explanation. In Stage 1, LLaVA-Mini learns to align vision and language representations using visual caption data. In Stage 2, LLaVA-Mini is trained to perform various visual tasks based on minimal vision tokens with one model. During this stage, we use visual SFT data to train LLaVA-Mini (with generic compression and modality pre-fusion) to handle a range of visual tasks. The entire pretraining-then-SFT process is similar to the setup in LLaVA-v1.5.
>
> Finally, only one model is needed to perform all visual tasks in our experiments. When handling different tasks, the compression module adaptively extracts key visual information from the input image and compresses it into a single vision token and the modality pre-fusion module fuses visual information into the text modality based on the input image and instruction (which may involve different tasks).
>
> We hope this clarification helps provide a clearer understanding of the LLaVA-Mini training process.
>
> &nbsp;
>
> **Q2**: Can we re-use the first few layers, which have high activation for vision tokens, as the fusion module? To further include the compression module, the first few layers can have two pathways, one pathway is to output the compressed tokens while the other pathway is to be gradually fused together with the text tokens.
>
> **A2**: Thank you for your valuable suggestion. We indeed considered the approach you proposed during the initial design phase of our method. However, two key limitations led us to not adopt it:
>
> - **Vision representations after the early layers contain contextual information, which complicates the compression module**: Once vision tokens are fed into the LLM, the early layers imbue the visual representations with contextual information. Applying query-based compression at this stage makes it difficult for the compression module to effectively distinguish between different vision tokens.
>
> - **Inter-layer operations within the LLM may not be compatible with existing acceleration frameworks**: One of the primary motivations for placing the compression and pre-fusion modules outside the LLM backbone in LLaVA-Mini is to keep the LLM backbone unchanged. This design choice ensures compatibility with a wide range of existing LLM acceleration technologies and frameworks, thereby further boosting efficiency.
>
> These considerations explain our architectural choices. According to your suggestion, **we have also included a comparison between LLaVA-Mini and a version of LLaVA-Mini with compression applied at the 4th layer of the LLM**. The results demonstrate that the current configuration of LLaVA-Mini offers more favorable performance. Following your suggestion, We have incorporated these result sand the rationale behind our architectural decisions into the *Appendix E.4  (Line 1134-1163)* in the new version of the paper.
>
> | Methods                                     | #Vision Tokens | FLOPs (T) | VQAv2 | GQA  | MMB  |
> | ------------------------------------------ | -------------- | --------- | ----- | ---- | ---- |
> | LLaVA-Mini                                 | 1              | 1.96      | 77.6  | 60.9 | 65.6 |
> | LLaVA-Mini (Two pathways: compress and fuse at 4-th layer in LLM) | 1              | 1.84      | 76.3  | 60.1 | 64.5 |

---

> > ### Comment · Reviewer_CwER · 2024-11-26
> >
> > I would like to thank the authors for the response. I remain very positive about this paper.

---

> ### Author Response · Authors · 2024-11-20
> **Response to Reviewer CwER (Part 2 of 2)**
>
> **Q3**: About missing reference of streaming video LLM works such as "VideoLLM-online: Online Video Large Language Model for Streaming Video. CVPR 2024"?
>
> **A3**: Thank you for your careful review and for suggesting the related work. VideoLLM-online is a valuable work, while it focuses more on similar generation with streaming videos, with different motivation and emphasis compared to LLaVA-Mini. In the new version, we will include a discussion of VideoLLM-online in the related work section.
>
> &nbsp;
>
> **Thanks again for your careful and valuable comments. Hope our responses answer your questions well and reassure your concerns.**

---

### Official Review · Reviewer_eQGe · 2024-10-30

**Soundness:** 3
**Presentation:** 3
**Contribution:** 2
**Rating:** 6
**Confidence:** 4

**Summary:**

a. The paper proposes LLaVA-Mini, an MLLM with improved efficiency via token compression. Specifically, the authors first unveil that early layers of MLLM are important for visual understanding. On top of this finding,  the efficient MLLM is then designed with 2 strategies: 1) compressing input visual tokens using the query-based resampler and 2) fusing visual tokens with text instruction tokens in early layers of LLM.The paper is well written, and delivered with clear motivation and simple yet effective methods.

b. The authors conducted comprehensive experiments to demonstrate the effectiveness of the methods. On the other hand, LLava-mini manages to achieve competitive performance with much-reduced tokens.

**Strengths:**

A. The paper is well written, and delivered with clear motivation and simple yet effective methods.

B. The authors conducted comprehensive experiments to demonstrate the effectiveness of the methods. On the other hand, LLava-mini manages to achieve competitive performance with much-reduced tokens.

**Weaknesses:**

A. Since the token numbers are substantially reduced. LLaVA-mini may achieve inferior performance under scenarios demanding fine-grained visual details, e.g., text-oriented benchmarks including Text VAQ, especially when compared with models with longer visual sequence input （e.g., LLaVA-Next)

B. The paper lacks an important baseline, namely, keeping the visual tokens in early layers of LLMs and then compressing the visual token output on the L-th (4 in the paper) layer.

**Questions:**

The authors claim that for video inputs, they train and infer via sampling frames at 1fps. In other words, the number of compressed visual tokens for videos of different lengths varies. I’d like to know if there’s any strategy (e.g, temporal embedding) to solve the discrepancy between training and inference (i.e., much longer videos).

---

> ### Author Response · Authors · 2024-11-20
> **Response to Reviewer eQGe (Part 1 of 2)**
>
> Thanks for your careful and valuable comments, and we will refine the paper following your suggestions. Following, we will respond to your questions in detail.
>
> &nbsp;
>
> **Q1**: May achieve inferior performance under scenarios demanding fine-grained visual details, e.g., text-oriented benchmarks including Text VQA.
>
> **A1**: Indeed, to achieve significant efficiency improvements, LLaVA-Mini compresses visual information, which entails a trade-off between efficiency and visual comprehension quality. Based on our experimental results, we believe that the slight performance decline on TextVQA is acceptable when weighed against the substantial efficiency gains. We appreciate your suggestion and have included a discussion of this limitation and potential  in the *Appendix H (Line 1345-1407)* of the new version.
>
> - **Flexible Adaptation to Different Visual Scenarios**: LLaVA-Mini must balance efficiency (the number of vision tokens) and performance (visual understanding capability). It allows for flexibility by adjusting the hyperparameter C, which controls the number of vision tokens, offering the possibility of adapting to different scenarios. Results in Table 1 and Table 6 demonstrate that performance improves consistently as C increases. In practice, for scenarios requiring fine-grained visual details, users can set a larger C to capture more visual information. Conversely, for scenarios where efficiency is prioritized, users can set a smaller C, achieving significant efficiency gains while retaining substantial visual understanding capability.
>
> - **Comparison with Previous Token Reduction Methods**: As you mentioned, most previous token reduction methods result in considerable loss of visual information. This serves as a core motivation for LLaVA-Mini, which seeks to maintain substantial visual comprehension while minimizing vision tokens through modality pre-fusion. The results in Table 1 further confirm that LLaVA-Mini outperforms prior approaches in terms of visual understanding.
>
> Thank you for your insightful comments. We hope these clarifications address your concerns. We have also incorporated this limitation and its corresponding solutions into the *Appendix H (Line 1345-1407)* in the new version of the paper.
>
> &nbsp;
>
> **Q2**: The baseline that keeps the visual tokens in early layers of LLMs and then compressing the visual token output on the L-th (4 in the paper) layer.
>
> **A2**: Thank you for your valuable feedback. This approach was considered during the early stages of our work; however, two key limitations led us to not adopt it:
>
> - **Vision representations after the L-th layers contain contextual information, which hinders the compression module**: After the vision tokens are fed into the LLM, the early layers cause the visual representations to carry contextual information. Applying query-based compression on top of these representations makes it difficult for the compression module to distinguish between different vision tokens.
>
> - **The inter-layer operations within the LLM may not be compatible with existing acceleration frameworks**: One of the main motivations for placing the compression and pre-fusion modules outside the LLM backbone in LLaVA-Mini is to keep the LLM backbone unchanged. This design allows for compatibility with nearly all existing LLM acceleration technologies and frameworks, further enhancing efficiency.
>
> The above points explain our architectural choices. Following your suggestion, **we have also included a comparison between LLaVA-Mini and LLaVA-Mini (with compression at the 4th layer in the LLM)**. The results demonstrate that the configuration of LLaVA-Mini is more advantageous. Following your suggestion, We have incorporated these results and the architectural motivation into the *Appendix E.4 (Line 1134-1163)* in the new version of the paper.
>
>
> | Methods                                    | #Vision Tokens | FLOPs (T) | VQAv2 | GQA  | MMB  |
> | ------------------------------------------ | -------------- | --------- | ----- | ---- | ---- |
> | LLaVA-Mini                                 | 1              | 1.96      | 77.6  | 60.9 | 65.6 |
> | LLaVA-Mini (compress at 4-th layer in LLM) | 1              | 1.84      | 76.3  | 60.1 | 64.5 |

---

> ### Author Response · Authors · 2024-11-20
> **Response to Reviewer eQGe (Part 2 of 2)**
>
> **Q3**: If there’s any strategy (e.g, temporal embedding) to solve the discrepancy between training and inference (i.e., much longer videos)?
>
> **A3**: The issue you raised is also of great interest to us. In LLaVA-Mini, the extension of training and inference capabilities, as well as the handling of longer video sequences, is achieved through the inherent context generalization capacity of the LLM itself, without the need for additional techniques. **The ability of LLaVA-Mini to effectively generalize in this manner stems from representing each image as a single token during video processing.** By using a single token per image, the position encoding implicitly facilitates the modeling of temporal information.
>
> Moreover, due to the compression of vision tokens, **the required generalization span for LLaVA-Mini is relatively small**. For example, the training process only involves 60 seconds (60 frames), while inference can involve up to 1 hour (3600 frames). For LLaVA-Mini, the context only needs to generalize from 60 tokens to 3600 tokens, which is considerably easier than LLaVA-v1.5, which must generalize from 60×576 tokens to 3600×576 tokens.
>
> Additionally, you mentioned that incorporating temporal embeddings could further enhance the model's explicit perception of temporal information. We plan to explore this in future work.
>
> &nbsp;
>
> **Thanks again for your careful and valuable comments to improve our work. If our responses answer your questions well and reassure your concerns, we would appreciate if you could reassess our work and increase the rating.**

---

> ### Author Response · Authors · 2024-11-30
> **Thanks for your valuable and insightful comments.**
>
> We sincerely appreciate your valuable and insightful comments, which have significantly contributed to the improvement of our work. **Hope that the additional experiments and responses we have provided dispel your concerns effectively and enhance the overall quality of the submission, potentially influencing your assessment rating of our work.**
>
> As the deadline for discussion (December 2nd) approaches, we would be grateful if you could let us know if there are any further questions and we are happy to respond as soon as possible.
>
> **Once again, we deeply appreciate the time and effort you have dedicated to the review and discussion process.**

---

### Official Review · Reviewer_WP7G · 2024-11-04

**Soundness:** 3
**Presentation:** 3
**Contribution:** 3
**Rating:** 6
**Confidence:** 4

**Summary:**

This paper presents LLaVA-Mini, an efficient large multimodal model that significantly reduces computational overhead while maintaining competitive performance. The key innovation lies in its ability to represent visual information with as few as one vision token, achieved through two main components: a modality pre-fusion module and a vision token compression module. The authors first conduct a layer-wise analysis of how LLMs process vision tokens, revealing that vision tokens are most crucial in early layers. Based on this insight, they design the pre-fusion module to process visual information before it enters the LLM, and employ a compression mechanism to minimize the number of vision tokens. Experimental results across 11 image and 7 video benchmarks demonstrate that LLaVA-Mini achieves comparable performance to LLaVA-v1.5 while reducing vision tokens from 576 to 1, resulting in 77% FLOPs reduction and significant memory savings (from 360MB to 0.6MB per image). This enables processing of long videos exceeding 10,000 frames and reduces inference latency from 100ms to 40ms.

**Strengths:**

1. Quality:
- Comprehensive empirical validation across 18 benchmarks (11 image-based and 7 video-based) demonstrates the robustness of the approach.
- The method achieves significant efficiency improvements (77% FLOPs reduction, 600x memory reduction) while maintaining competitive performance with the baseline.

2. Clarity:
- The motivation and problem formulation are well-articulated, clearly identifying the computational challenges in current LMMs.
- The paper presents a logical progression from empirical observations to proposed solutions.

3. Significance:
- The dramatic reduction in computational resources makes multimodal processing more accessible.
- The approach addresses a critical limitation in current LMMs, potentially enabling real-time applications and processing of high-resolution inputs.

**Weaknesses:**

1. Writing:
- The attention weight analysis lacks crucial technical details. The authors do not specify how they handle multiple attention heads (whether through averaging, individual analysis, or selective visualization), making the results difficult to reproduce.
- The paper fails to clarify how attention scores are averaged among instruction, vision, and response tokens, which is essential for understanding the analysis methodology.
- Despite using LLaVA 1.5 and LLaVA NeXT as base models, which both incorporate high and low-resolution vision inputs, the analysis does not differentiate between these different types of vision tokens.

2. Technical Inconsistencies:
- There's a discrepancy between the visualization results and claims: while the authors argue that attention to vision tokens decreases in latter layers, the entropy visualization shows persistently high values for vision tokens in these layers, creating confusion about the actual attention distribution.
- The paper lacks sufficient details about the pre-fusion transformer architecture and implementation, which is crucial for reproduction and understanding the method's effectiveness.

3. Architectural Design and Claims:
- The paper's title and main claim about "one vision token" is potentially misleading. The implementation still requires processing full projected vision tokens in the pre-fusion transformer, suggesting that the computational efficiency gains are not solely from token reduction.
- The ablation studies (lines 446-457) emphasize the importance of the pre-fusion module, raising questions about the actual contribution of the token compressor. The paper should explore the effectiveness of using only the pre-fusion module without compression.

4. Computational Analysis:
- The paper lacks a detailed analysis of the computational overhead introduced by the fusion module, particularly regarding the processing of N^2×5 vision tokens plus l_q text tokens.
- The training costs during the instruction tuning stage, including memory and FLOPs comparisons with LLaVA, are not adequately addressed, making it difficult to assess the true efficiency gains during training.

5. Logic:
- The presentation of "Modality Pre-fusion" and "Vision Token Compression" would benefit from a chronological reorganization to better reflect the processing pipeline (image encoder -> vision token compression -> modality pre-fusion).

**Questions:**

See "Weakness".

Additional question:

1. Entropy Visualization Clarification:
- Could the authors explain the apparent contradiction between the claim of decreased attention to vision tokens in the latter layers and the high entropy values shown in the visualization?
- What causes high entropy in vision-related attention weights in the latter layers despite lower attention weights?

---

> ### Author Response · Authors · 2024-11-20
> **Response to Reviewer WP7G (Part 1 of 3)**
>
> Thanks for your careful and valuable comments, and we will refine the paper following your suggestions. Following, we will respond to your questions in detail.
>
> &nbsp;
>
> **Q1**: About the writing of detailed settings in attention weight analysis.
>
> **A1**: We greatly appreciate your valuable suggestions. Due to space limitations, some specific details regarding the attention weight analysis were not included in the main body of the paper. Following your suggestion, we have provided a more detailed description of the analysis setup and formalized calculations in *Appendix A (Line 918-954)* .
>
> - **How to handle multiple attention heads**: The attention weights are averaged across all attention heads.
>
> - **How attention scores are averaged among instruction, vision, and response tokens**: Taking the attention scores from response to vision as an example, we compute the sum of attention weights from each response token to all vision tokens. Then, the attention from the response tokens to the vision tokens is averaged across all response tokens to obtain the average attention of response-type tokens to vision-type tokens. We have formalized this process and added the relevant details in Appendix A (Line 918-954).
>
> - **How to handle LLaVA 1.5 and LLaVA-NeXT with different resolutions**: For LLaVA-v1.5 (pad) and LLaVA-NeXT (anyres), which may involve vision inputs at different resolutions, we use the original settings as specified. In our analysis, we do not distinguish between different types of vision tokens but treat them collectively as vision tokens, measuring their overall importance.
>
> Thanks for your suggestion, **we have added a detailed description of the experimental setup for the attention analysis in Appendix A (Line 918-954)**. Additionally, after the paper is de-anonymized, we will **release the attention analysis scripts to ensure the reproducibility of our analysis**. Hope the added formal details and future open source code can dispel your concerns about the reproducibility of analysis.
>
> &nbsp;
>
> **Q2**: About the discrepancy between the visualization results and claims of attention to vision tokens decreases in later layers?
>
> **A2**: We apologize for any confusion caused.
>
> Figure 2 illustrates the attention weights received by vision tokens at each layer, which remain at very low levels in the later layers. The visualization in Figure 4 may have caused some misunderstanding, as it appears that instruction tokens still maintain some attention to vision tokens (appearing brighter) in the later layers. In reality, Figure 4 was designed to more clearly demonstrate the variation of attention distributions across different layers. Due to the overall sequence length, the attention scores for each position are quite small. Furthermore, the first token in the upper-left corner always receives 100% attention weight, which results in a visualization where only the upper-left position is bright, while the others appear extremely dark (since their values are very small). To make the visualization more readable and provide a clearer impression to the readers, **the color bar in Figure 4 is not evenly distributed but follows a logarithmic scale**. Thus, **although some vision tokens appear a little bit bright in the later layers of the visualization, when considered alongside the color bar, their attention weights are indeed very low**. Meanwhile, in the later layers of the visualization, the number of vision tokens receiving attention decreases significantly (with some columns of vision tokens appearing very dark), resulting in the overall attention weights for vision tokens remaining very low in these layers.
>
> Thank you for your reminder. In the new version of the paper, we have highlighted the uneven distribution of the color bar in the visualization (Line 196).
>
> &nbsp;
>
> **Q3**: Lacks sufficient details about the pre-fusion transformer architecture.
>
> **A3**: Thank you for your insightful feedback. The pre-fusion module applies **the same decoder-only architecture as the LLM, including both the structure and hyperparameters**. The motivation behind this setting is to ensure flexible adaptation to existing LLM frameworks and other acceleration techniques. Based on your suggestion, we have included a more detailed explanation of pre-fusion transformer in the *Appendix B (Line 958-967)* of the new version.

---

> ### Author Response · Authors · 2024-11-20
> **Response to Reviewer WP7G (Part 2 of 3)**
>
> **Q4**: About the claim of "one vision token" and the computational efficiency gains are not solely from token reduction.
>
> **A4**: Yes, the proposed LLaVA-Mini still processes all the projected vision tokens. The statement regarding "one vision token" refers to **the number of vision tokens fed into the LLM**, as the parameters and computational cost of the LLM backbone dominate the overall resource consumption in LMMs. Reducing the number of tokens fed into the LLM backbone can significantly enhance efficiency.
>
> This is also the key innovation of LLaVA-Mini compared to previous works. Token reduction methods, such as PruMerge and MQT-LLaVA, focus on directly reducing the number of tokens output by the ViT. In contrast, LLaVA-Mini focuses on transmitting visual information to the LLM using the fewest possible vision tokens, thereby enhancing efficiency while maintaining strong visual understanding.
>
> Thank you for your suggestion. In the new version, we have emphasized that "one vision token" refers specifically to the vision tokens fed into the LLM to avoid any potential misunderstanding.
>
> &nbsp;
>
> **Q5**: About the ablation study of using only the pre-fusion module without compression.
>
> **A5**: Thank you for your valuable suggestion. We have conducted the relevant ablation experiments in following table. As shown in table, when using only the pre-fusion module without compression, LLaVA-Mini achieves superior performance compared to LLaVA-v1.5 with both using 576 vision tokens, demonstrating the effectiveness of the pre-fusion module. We have included the experimental results in the *Appendix E.3 (Line 1120-1133)* in the new version of the paper.
>
> | Methods    | #Vision Tokens | VQAv2 | GQA   | MMB  |
> | ---------- | -------------- | ----- | ----- | ---- |
> | LLaVA-v1.5 | 576            | 78.5  | 62.0  | 64.3 |
> | LLaVA-Mini (w/o compression) | 576            | 80.0  | 62.9  | 66.2 |
>
> &nbsp;
>
> **Q6**: About the computational overhead introduced by the fusion module, particularly regarding the processing of N^2×5 vision tokens plus l_q text tokens.
>
> **A6**: We apologize for any confusion regarding the efficiency analysis in Section 5.3. The computational overhead reported in Figures 7 and 8 reflects the total computation required by the entire LLaVA-Mini model, including the vision encoder, compression module, pre-fusion module, and LLM backbone. Therefore, the computational overhead introduced by the fusion module is already accounted for in both Figures 7 and 8. In comparison, LLaVA-Mini demonstrates a clear efficiency advantage over LLaVA-v1.5.
>
> Regarding the processing of N^2×5 vision tokens, the results in Figure 8 further highlight the efficiency of LLaVA-Mini. Specifically, LLaVA-v1.5 requires the LLM to process over 2000 vision tokens for high-resolution images, while LLaVA-Mini’s pre-fusion module consists of only 4 layers, offering a significant reduction in computational cost.
>
> Additionally, **we believe you may be interested in the proportion of computational load contributed by the pre-fusion module in LLaVA-Mini**. To further study the proportion of computational load contributed by each component in LLaVA-Mini, we computed the FLOPs of each module, as shown in the table below. The proposed compression module and pre-fusion module incur minimal computational cost, while the computation required by the LLM backbone is significantly reduced. We have included the experimental results in the *Appendix E.6  (Line 1182-1187)* in the new version of the paper.
>
> | Methods    | Res. | Vision Encoder | Projection | Compression | Pre-fusion | LLM    | Total (TFLOPs) |
> | ---------- | ---- | -------------- | ---------- | ----------- | ---------- | ------ | -------------- |
> | LLaVA-v1.5 | 336  | 0.349          | 0.024      | -           | -          | 8.177  | 8.55           |
> | LLaVA-Mini | 336  | 0.349          | 0.024      | 0.001       | 0.125      | 1.460  | 1.96           |
> | LLaVA-v1.5 | 672  | 1.745          | 0.121      | -           | -          | 38.623 | 40.49          |
> | LLaVA-Mini | 672  | 1.745          | 0.121      | 0.009       | 1.183      | 4.131  | 7.19           |

---

> ### Author Response · Authors · 2024-11-20
> **Response to Reviewer WP7G (Part 3 of 3)**
>
> **Q7**: The presentation of "Modality Pre-fusion" and "Vision Token Compression" would benefit from a chronological reorganization to better reflect the processing pipeline (image encoder -> vision token compression -> modality pre-fusion).
>
> **A7**: Thank you for your thoughtful suggestion regarding the writing sequence. In fact, the modality pre-fusion is also computed based on the vision representations output by the vision encoder. **Logically, the compression module and the pre-fusion module in a parallel relationship**, which is why we have organized the paper in this way. We appreciate your feedback, and in the next version, we will provide a more detailed explanation in the method overview part and consider adjusting the structure to help readers better understand the overall framework.
>
> &nbsp;
>
> **Q8**: The training costs during the instruction tuning stage are not adequately addressed, making it difficult to assess the true efficiency gains during training.
>
> **A8**: The motivation behind LLaVA-Mini is to **improve the inference efficiency** of LMMs by reducing the number of vision tokens input to the LLM backbone. Training efficiency is not the focus of our research, because LLaVA-Mini is not designed to improve training efficiency, and we do not claim that it can improve training efficiency. We apologize for any misunderstanding caused and will emphasize this point more clearly in the next version.
>
> &nbsp;
>
> **Q9**: About the relation between decreased attention weight and high attention entropy? What causes high entropy in vision-related attention weights in the latter layers despite lower attention weights?
>
> **A9**: We apologize for any confusion caused.
>
> Firstly, we did not claim that there is high entropy in the vision-related attention weights in the later layers. As shown in Figure 3, we emphasize high entropy in the early layers (lines 186-188). In the later layers, however, **no consistent entropy patterns in the later layers are observed across different models**. For instance, LLaVA-v1.5-Vicuna-7B exhibits higher vision-related entropy in the later layers, while LLaVA-NeXT-Vicuna-7B shows lower entropy.
>
> **Additionally, attention weights and attention entropy are not directly correlated.**
>
> - **Attention weight**: The attention weight in Figure 2 measures how much attention a vision token receives from subsequent tokens. For example, when evaluating the attention of a response token towards previous tokens, the response token may attend to instruction tokens, vision tokens, and response tokens (prefix), with their attention weights summing to 1. Therefore, a decreased attention weight to vision tokens indicates a reduced importance of vision tokens in the later layers.
>
> - **Attention entropy**: The attention entropy in Figure 3 measures the degree of focus that vision tokens (note that the attention received by vision tokens is normalized to adapt to the calculation of entropy) receive from subsequent tokens. For instance, if the response token equally attends to all vision tokens, the entropy is high. Conversely, if the response token focuses on only a few vision tokens, the entropy is low.
>
> Thus, the observed decrease attention weight and attention entropy results is not contradictory. In the later layers, vision tokens receive less attention weight, and this attention could either be focused on a few vision tokens (low entropy) or spread across many vision tokens (high entropy).
>
> We hope this clarifies your question regarding attention entropy.
>
> &nbsp;
>
> **Thanks again for your careful and valuable comments to improve our work. If our responses answer your questions well and reassure your concerns, we would appreciate if you could reassess our work and increase the rating.**

---

> > ### Comment · Reviewer_WP7G · 2024-11-26
> >
> > Thank you for your engagement during the discussion phase. While many of my concerns have been addressed, I recommend a major revision to improve the clarity of the technical descriptions. Upon such improvements in the manuscript's writing, I would gladly consider raising my rating.

---

> > > ### Author Response · Authors · 2024-11-27
> > > **We are happy that our response have addressed many of your concerns! We have further improved the manuscript's writing based on your suggestion.**
> > >
> > > We are pleased that our response have addressed many of your concerns. Based on your feedback, we have further refined the technical descriptions and have uploaded the updated version of the paper.
> > >
> > > The improvements in the latest version include the following (marked in red):
> > >
> > > - **More Detailed Introduction of Preliminary Analyses**: In response to your suggestion, we have added formal representations of relevant statistics from the Preliminary Analyses section in Appendix A to enhance reproducibility. Due to space constraints in the under-review version, this section has been placed in the appendix. Given its importance, we plan to move it to the main text in the de-anonymized version.
> > >
> > > - **Improved Logical Flow of Technical Descriptions**: Thank you for your suggestion. We have re-reorganized the structure in Section 4 for better clarity: vision encoder → vision token compression → modality pre-fusion. Additionally, we have included a more detailed overview at the beginning of Section 4 to provide readers with a clearer understanding of the overall LLaVA-Mini framework.
> > >
> > > - **Additional Experiments**: As per your request, we have included relevant ablation studies in Appendix E to help readers better understand the contributions of each module.
> > >
> > > - **More Detailed Descriptions**: Following your feedback, we have added specific details about various components in the Technical Descriptions section, such as the structure and parameters of the pre-fusion module.
> > >
> > > Once again, we greatly appreciate your valuable suggestions to enhance our work. We hope the latest version has resolved your concerns we would appreciate if you consider raising the rating.

---

### Official Review · Reviewer_rV2Y · 2024-11-06

**Soundness:** 3
**Presentation:** 3
**Contribution:** 3
**Rating:** 6
**Confidence:** 4

**Summary:**

LLaVA-Mini proposes a compute and memory efficient Multi-modal by compressing the vision tokens that is passed to the LLM. The work achieves this by introducing a technique of modality pre-fusion where visual representation is fused into the text tokens and compression of vision tokens using cross-attention with learnable compression queries. LLaVA-Mini achieves comparable performance to existing models like LLaVA-v1.5 across several benchmarks, while significantly improving efficiency in terms of FLOPs, latency, and memory usage.

**Strengths:**

- The finding that vision tokens are primarily crucial in early layers is key to the design of the efficient early fusion architecture, a significant contribution to reducing computational overhead in LLMs.
- Overall, the paper is well written and easy to follow. The methods, concepts are well explained and tables/ figures convey key findings. The qualitative visualization examples such as the cross-attention in compression module helps provide intuition.
-   LLaVA-Mini achieves significant performance, reducing FLOPs by 77% and VRAM usage by more than 99% per image compared to LLaVA-v1.5, without compromising much in terms of model accuracy. This efficiency could make LLaVA-Mini highly suitable for edge devices, making it as a practical solution for real-time multimodal interactions.
- The authors provide exhaustive ablations experiments on number of pre-fusion layers/ vision tokens etc.

**Weaknesses:**

- The paper does not explain the complexity of plugging this compression method to existing multi-modal pipelines. The lack of open source implementation of the code might hinder reproducibility.
- While the paper highlights successful qualitative examples, it would be valuable if it could also discuss any failure cases encountered with the compression technique, along with potential solutions for addressing them.
- It would be great to understand how model efficiency scales across different hardware platforms.

**Questions:**

- Have the authors done any ablation on using SOTA token pooling methods for the vision token pooling and compare against them (other than average pooling).
- Can the authors highlight results on more vision centric benchmarks and show the impact of different number of vision tokens on these e.g., CV-Bench [1]





[1] Cambrian-1: A Fully Open, Vision-Centric Exploration of Multimodal LLMs

---

> ### Author Response · Authors · 2024-11-20
> **Response to Reviewer rV2Y (Part 1 of 2)**
>
> Thanks for your careful and valuable comments, and we have refined the paper following your suggestions. Below, we will respond to your questions in detail.
>
> &nbsp;
>
> **Q1**: About the complexity of plugging this compression method to existing multi-modal pipelines, and the open source implementation of the code.
>
> **A1**: Thank you for your suggestion. The compression method of LLaVA-Mini can be easily plugged into existing multi-modal pipelines, as it **only requires the addition of two extra modules (the compression module and the modality pre-fusion module) before the LLM**, while the other components (such as the vision encoder, the LLM, and the training loss) remain unchanged. Following your advice, we have includes a description of how to integrate the compression method into existing multi-modal pipelines in the *Appendix B (Line 958-967)* in the new version.
>
> More importantly, we have **provided a source code of LLaVA-Mini in the supplementary materials and plan to open-source the GitHub repository** after the paper is anonymized, to ensure the reproducibility of LLaVA-Mini.
>
> Hope the additional description and code in the supplementary materials can dispel your concerns about reproducibility.
>
> &nbsp;
>
> **Q2**: About the limitation of the compression technique and potential solutions for addressing them?
>
> **A2**: Thank you for your valuable suggestion. We have explained the limitation of LLaVA-Mini in the *Appendix H (Line 1345-1407)*  of the new version. The limitation of LLaVA-Mini lies in the trade-off between the number of vision tokens (efficiency) and performance. As LLaVA-Mini uses a single vision token, it compresses visual information, which may leave room for improvement in image understanding tasks that involve complex visual content. A potential solution is to slightly increase the number of vision tokens, such as using 16 vision tokens, to achieve a better balance between efficiency and performance. Query-based compression offers flexibility in adjusting the number of vision tokens, as we can simply modify the parameter C to control the number of tokens. The results in Table 6 and Table 8 demonstrate that increasing the number of vision tokens leads to continuous performance improvements in LLaVA-Mini.
>
> In practice, the number of vision tokens can be adjusted based on the specific efficiency requirements of different scenarios, allowing for a trade-off between efficiency and performance.  We appreciate your feedback, and have incorporated the limitation and potential solutions of LLaVA-Mini into the *Appendix H (Line 1345-1407)* in the new version of the paper.
>
> &nbsp;
>
> **Q3**: How model efficiency scales across different hardware platforms?
>
> **A3**: Thank you for your reminder. The efficiency improvements brought by LLaVA-Mini stem from reduced computational load (FLOPs), which is consistent across different hardware platforms. To demonstrate the scalability of model efficiency across different hardware platforms, we have added experiments that compute the inference latency of LLaVA-Mini on three hardware configurations: RTX 3090, A100, and A800, as presented in the table below. The results indicate that the efficiency improvements brought by LLaVA-Mini are scalable across these platforms.
>
> | Latency (ms)                   | RTX 3090 | A100   | A800  |
> | ------------------------------ | -------- | ------ | ----- |
> | LLaVA-v1.5 (576 vision tokens) | 198.75   | 113.04 | 87.43 |
> | LLaVA-Mini  (1 vision tokens)  | 64.52    | 38.64  | 27.43 |
> | LLaVA-Mini  (4 vision tokens)  | 65.52    | 38.84  | 27.71 |
> | LLaVA-Mini  (16 vision tokens) | 68.97    | 39.28  | 28.92 |
> | LLaVA-Mini  (64 vision tokens) | 80.10    | 46.23  | 34.65 |
>
> Following your recommendation, we have included the this experiment in *Appendix E.5 (Line 1165-1180)* of the new version.

---

> ### Author Response · Authors · 2024-11-20
> **Response to Reviewer rV2Y (Part 2 of 2)**
>
> **Q4**: Have the authors done any ablation on using SOTA token pooling methods for the vision token pooling and compare against them (other than average pooling)?
>
> **A4**: Actually, by examining both Table 1 and Table 6 in the paper together, we can compare the query-based compression method proposed by LLaVA-Mini with state-of-the-art (SOTA) token pooling methods. **We appreciate your suggestion and have included an ablation study in the tables that compares LLaVA-Mini with various token pooling methods.** To ensure a fair comparison of token compression performance, we have removed the modality pre-fusion module from LLaVA-Mini for the comparison with SOTA token merging methods, including PruMerge, PruMerge++, and MQT-LLaVA. Specifically, PruMerge applies the widely-used token merge (ToMe) technique on ViT, PruMerge++ improves upon PruMerge by uniformly sampling additional vision tokens, and MQT-LLaVA employs Matryoshka representation learning to compress vision tokens.
>
> | Methods    | #Vision Tokens | VQAv2 | GQA  | MMB  |
> | ---------- | -------------- | ----- | ---- | ---- |
> | MQT-LLaVA  | 2              | 61.0  | 50.8 | 54.4 |
> | MQT-LLaVA  | 36             | 73.7  | 58.8 | 63.4 |
> | MQT-LLaVA  | 256            | 76.8  | 61.6 | 64.3 |
> | PruMerge   | 32             | 72.0  | -    | 60.9 |
> | PruMerge++ | 144            | 76.8  | -    | 64.9 |
> | LLaVA-Mini | 1              | 72.4  | 54.2 | 57.7 |
> | LLaVA-Mini | 16             | 74.1  | 55.4 | 59.2 |
> | LLaVA-Mini | 64             | 75.3  | 56.7 | 62.1 |
> | LLaVA-Mini | 144            | 76.9  | 58.9 | 64.9 |
>
>
> As shown in the table, LLaVA-Mini's compression module outperforms PruMerge, PruMerge++, and MQT-LLaVA at the same compression rate, showing the advantages of query-based compression. Following your valuable suggestion, we have added this detailed comparison between query-based compression and SOTA token pooling methods in the *Appendix E.2 (Line 1094-1119)* of the new version.
>
> &nbsp;
>
> **Q5**: Can the authors highlight results on more vision centric benchmark and show the impact of different number of vision tokens on these e.g., CV-Bench?
>
> **A5**: Thank you for your valuable suggestion. Evaluating LLaVA-Mini on vision-centric benchmarks would further demonstrate its understanding capabilities of visual information. **CV-Bench is an excellent benchmark for assessing vision-centric capabilities, encompassing both 2D and 3D visual understanding**. In the table below, we compare LLaVA-Mini with LLaVA-v1.5, where LLaVA-Mini demonstrates superior vision-centric understanding with fewer vision tokens on CV-Bench.
>
> Following your suggestion, we have included these experimental results in the *Appendix E.1 (Line 1077-1092)* of the new version.
>
>
> | Methods    | #Vision Tokens | CVBench-2D | CVBench-3D | Avg.  |
> | ---------- | -------------- | ---------- | ---------- | ----- |
> | LLaVA-v1.5 | 576            | 61.96      | 58.58      | 60.27 |
> | LLaVA-Mini | 1              | 62.31      | 69.33      | 65.82 |
> | LLaVA-Mini | 4              | 63.42      | 72.00      | 67.71 |
> | LLaVA-Mini | 16             | 65.58      | 73.75      | 69.66 |
>
> &nbsp;
>
> **Thanks again for your careful and valuable comments to improve our work. If our responses answer your questions well and reassure your concerns, we would appreciate if you could reassess our work and increase the rating.**

---

> ### Author Response · Authors · 2024-12-02
> **Thanks for your valuable and insightful comments.**
>
> We sincerely appreciate your valuable and insightful comments, which have significantly contributed to the improvement of our work. Hope that the additional experiments and responses we have provided dispel your concerns effectively and enhance the overall quality of the submission, potentially influencing your assessment rating of our work.
>
> As the deadline for discussion (December 2nd) approaches, we would be grateful if you could let us know if there are any further questions and we are happy to respond as soon as possible.
>
> Once again, we deeply appreciate the time and effort you have dedicated to the review and discussion process.

---

### Author Response · Authors · 2024-11-24
**Thanks all reviewers for their valuable comments!**

We sincerely thank the reviewers for their recognition of our work and for providing thoughtful and constructive comments that have significantly contributed to improving our work. Based on the reviewers’ suggestions, we have carefully revised and strengthened the paper.

Since the public discussion phase nears its conclusion, if the reviewers have any additional questions, we would be delighted to address them promptly.

Once again, we are deeply grateful for the reviewers’ insightful comments and support.

---

### Public Comment · ~Liang_Chen10 · 2025-03-05
**Reporting Academic Plagiarism in an Accepted ICLR 2025 Paper (LlavaMini, Submission Number: 13752)**

Dear ICLR 2025 Program Chair, AC, Reviewers of LlavaMini

I am writing to report a serious case of academic plagiarism in a paper accepted for ICLR 2025. The paper titled "LlavaMini" has substantially plagiarized my work "FastV, https://arxiv.org/abs/2403.06764" which was published at ECCV 2024.

Upon thorough examination of the LlavaMini paper, I have identified clear instances where my methodologies, technical approaches, and results have been appropriated without proper attribution. LlavaMini even does not cite FastV given so many similarities.  Most notably:

As documented in a public GitHub issue (https://github.com/ictnlp/LLaVA-Mini/issues/14#issuecomment-2692746529), Figure 4 in LlavaMini is strikingly similar to Figure 4 in FastV, featuring identical token nomenclature and even the same color scheme. The resemblance is so precise that it strongly suggests use of our codebase. What's particularly concerning is that LlavaMini presents these elements—specifically the finding that "Vision Tokens are More Important in Early Layers"—as novel contributions, when this was in fact a key insight published in our work eight months ago.

I find it troubling that all reviewers and the Area Chair failed to identify this plagiarism, especially considering that FastV has garnered over 100 citations in the community since its publication. A paper should not be accepted when one of its main claimed contributions is directly taken from prior published work.

I have attempted to contact the authors of LlavaMini regarding this matter but have not received any direct response. As a researcher who regularly submits to and reviews for ICLR, I believe this issue must be addressed seriously to maintain the integrity and reputation of the conference.

Thank you for your attention to this serious matter. I am available to provide any additional information or evidence required for your investigation.

---

> ### Public Comment · ~Shaolei_Zhang1 · 2025-03-08
> **Response to the False Claim (PART 1/2)**
>
> Dear ICLR 2025 Program Chair, Liang Chen
>
> Upon receiving his comments and email, we were surprised, as it was completely unexpected.
>
> First, we would like to state that, **prior to receiving an email from Liang Chen on March 2, 2025, we were unaware of the existence of FastV**. This is because FastV was not mentioned in any of our highly-relevant baselines (studies focused on compressing the number of vision tokens fed into the LLM backbone, such as LLaVA-PruMerge, VoCo-LLaMA, MQT-LLaVA, and TokenPacker).
>
> On March 2, 2025, we received the first email from Liang Chen, in which he pointed out that one of the three observations in our preliminary analysis—"Vision Tokens Are More Important in Early Layers"—aligned with their findings, and he suggested that we cite FastV. *Upon reviewing this, we promptly included the citation in the latest version and replied to his email accordingly*.
>
> On March 3, 2025, Liang Chen responded with additional requests, asking us to:\
> 1.	Remove the claim "To this end, we begin by exploring a foundational question: How does the LMM (particularly the LLaVA architecture) understand vision tokens?" Replace it with "Inspired by FastV, we conduct the same experiments..."\
> 2.	Acknowledge FastV's prior work at the beginning of this section.\
> 3.	Include a comparison with FastV's methods.\
> 4.	Explicitly discuss the differences between FastV and your work.
>
> From our perspective, **these requests were inappropriate and exceed reasonable boundaries**. Since we had not read their paper before, the demands to include statements such as "Inspired by FastV..." and "acknowledgment..." were factually inaccurate. Furthermore, LLaVA-Mini differs fundamentally from FastV in both methodology and conclusions. **LLaVA-Mini introduces a novel architecture and training paradigm, whereas FastV is an inference method. LLaVA-Mini processes vision tokens extracted by the vision encoder, whereas FastV reduces the number of vision tokens at the intermediate layers of the LLM. The two works were evaluated on almost different benchmarks, and there are basically no identical experimental results.** Given these fundamental differences on method and innovation, a direct comparison with FastV exceeds reasonable requests, just as none of the related baselines we compared against had referenced FastV. Since these requests were beyond a reasonable scope, we chose not to address them further. Subsequently, we received the email that Liang Chen posted on the OpenReview platform.

---

> ### Public Comment · ~Shaolei_Zhang1 · 2025-03-08
> **Response to the False Claim (PART 2/2)**
>
> > **LLaVA-Mini and FastV differ entirely in method and results.**
>
> Liang Chen pointed out that a part of our preliminary experiments reached the same observation as their study. However, this similarity is limited to observational findings and does not pertain to the core methodology or innovations of our work.
>
> The primary contribution and innovation of LLaVA-Mini lie in its ability to compress vision tokens fed into the LLM backbone into a single token through the proposed learnable compression module and modality pre-fusion module. We had not read their paper beforehand and independently arrived at a similar observation under unknown circumstances. Moreover, this observation constitutes only a minor part of LLaVA-Mini and is not central to our main contribution, and we have never claimed that we are the first to discover it. Therefore, we have appropriately cited their work in this observation, while the accusation of plagiarism is both unwarranted and excessive.
>
> > **It is highly groundless and inappropriate to make accusations of plagiarism based solely on the use of a similar color scheme in attention visualization.**
>
> Liang Chen’s concerns stem from the situation that our **attention visualizations use the same color scheme as theirs**, which led to the claim that “features the identical token nomenclature and even the same color scheme, suggests we use their codebase.” **It is inconceivable that a similar color scheme in visualization could serve as evidence for accusing us of using their code.**
>
> Using this as a basis to falsely accuse us of plagiarizing the FastV codebase is entirely unfounded. The token nomenclature in our work follows the naming conventions established in the original LLaVA paper, along with the corresponding token positions. As for the color scheme, we simply used cmap='viridis', a most widely used setting for attention visualization. Attention visualization is one of the most fundamental analytical techniques, and therefore, the explicit speculation that we used FastV’s codebase is entirely unacceptable.
>
> In principle, we are not obligated to prove the originality of our code, as the allegation itself is baseless and purely speculative. However, since ICLR comments are publicly visible, such unfounded accusations have caused serious reputational harm to our work. Demonstrating the exact process by which code is created is not always straightforward. Fortunately, in this case, the complete visualization code was generated entirely from scratch using ChatGPT via textual prompts, with a recorded timestamp of August 30, 2024 (https://chatgpt.com/share/67c817bb-0af0-8013-887c-e022c5c04532). **This directly invalidates the premise of Liang Chen’s claim that we used their codebase.**
>
> &nbsp;
>
> We appreciate ICLR for providing a platform for open academic exchange. **We believe that it is unreasonable to make overly subjective accusations of plagiarism based solely on an attention visualization image with a similar color scheme, especially when the paper’s writing, model, training, and experimental results are entirely different.** While OpenReview is an open platform for academic discussion, this does not allow unwarranted and exaggerated accusations, which run counter to the ICLR community's core values of fairness, mutual respect, and constructive exchange. Such speculation has unfairly impacted our work, and we request a clarification.
>
> We sincerely appreciate the Program Chair’s time and effort on this matter.

---

> ### Public Comment · ~Liang_Chen10 · 2025-03-12
> **Unawareness Is Not a Vaild Excuse**
>
> Dear ICLR 2025 Program Chair, Shaolei and All,
>
> I must point out that claiming "prior to receiving an email from Liang Chen on March 2, 2025, we were unaware of the existence of FastV" is not a valid excuse for several reasons:
>
> - Unawareness itself is not verifiable. **In fact** we have evidence indicating LLaVA-Mini team was familiar with FastV as early as March 2024 (1 year ago). We can show the evidence if it is required by PC.
> - Failing to acknowledge highly related work constitutes a significant oversight. One of the baselines you cited (https://openreview.net/forum?id=gZnBI7WS1K) was rejected by ICLR 2025 specifically for "unawareness of FastV." according to the reviewers. It is inequitable that the same oversight is treated differently in this case.
> - Even by the author's own description of LLaVAMini ("LLaVA-Mini processes vision tokens extracted by the vision encoder, whereas FastV reduces the number of vision tokens at the intermediate layers of the LLM"), the close relationship between the two approaches is so evident that your claim "LLaVA-Mini and FastV differ entirely in method and results" seems highly questionable and difficult to reconcile with the technical details.
>
> Thank you for your attention to this serious matter.

---

### Meta-Review · Area_Chair_7nsR · 2024-12-20

**Metareview:**

This work presented a novel way of compressing visual tokens for images and videos in multimodal LLMs. The authors started with an analysis on how the LLMs handle visual tokens in the middle layers and found that most tokens only play important roles at eearly stages. Based on this observation, the authors proposed a new visual token compression algorithms to conduct pre-fusion for visual tokens before feeding them into LLMs. As a result, the model can support down to one visual tokens for LLMs while still retaining a great performance across variou multimodal benchmarks.

The main strength of this work, as pointed out by the reviewers, is that the authors proposed a new way of compressing long sequence of visual tokens into a smaller one for multimodal LLMs. The authors did thorough analysis on how LLMs cope with visual tokens, from which they derived a neat approach to compress the visual tokens by modality pre-fusion. The authors further showed solid improvement of efficiency while reserving the effectiveness. The discussions between authors and reviewers are pretty engaged.

A few weaknesses were pointed out by the reviewers, including the paper writing, missed report of complexity for the fusion module, etc. After the rebuttal, the authors mostly addressed the concerns and revised the submission accordingly. One more concern about this work is whether this work applies for those tasks which require fine-grained information, such as DocVQA, as pointed out by Reviewer eQGe. The ACs believe this part requires more study for the proposed method.

Given that all reviewers gave positive rating to this work, and most of the concerns have been addressed by the authors during rebuttal, the ACs recommend an acceptance to this work.

**Additional Comments On Reviewer Discussion:**

The reviewer discussions are of much engagement between the authors and reviewers, which significantly help to resolve any misunderstandings during the review stages. Finally, all reviewers reached a consensus about the positive impact of this work to the community.

---

### Decision · Program_Chairs · 2025-01-22

Accept (Poster)